# Integration of Tmc1/2 into the mechanotransduction complex in zebrafish hair cells is regulated by Transmembrane O-methyltransferase (Tomt)

Timothy Erickson[1], Clive P Morgan[1], Jennifer Olt[2], Katherine Hardy[2], Elisabeth Busch-Nentwich[3], Reo Maeda[1], Rachel Clemens[1], Jocelyn F Krey[1], Alex Nechiporuk[4], Peter G Barr-Gillespie[1], Walter Marcotti[2], Teresa Nicolson[1]*

[1]Oregon Hearing Research Center and the Vollum Institute, Oregon Health and Science University, Portland, United States; [2]Department of Biomedical Science, University of Sheffield, Sheffield, United States; [3]Wellcome Trust Sanger Institute, Cambridge, United Kingdom; [4]Department of Cell, Developmental and Cancer Biology, Oregon Health and Science University, Portland, United States

*For correspondence: nicolson@ohsu.edu

Competing interests: The authors declare that no competing interests exist.

**Abstract** Transmembrane O-methyltransferase (*TOMT/LRTOMT*) is responsible for non-syndromic deafness DFNB63. However, the specific defects that lead to hearing loss have not been described. Using a zebrafish model of DFNB63, we show that the auditory and vestibular phenotypes are due to a lack of mechanotransduction (MET) in Tomt-deficient hair cells. GFP-tagged Tomt is enriched in the Golgi of hair cells, suggesting that Tomt might regulate the trafficking of other MET components to the hair bundle. We found that Tmc1/2 proteins are specifically excluded from the hair bundle in *tomt* mutants, whereas other MET complex proteins can still localize to the bundle. Furthermore, mouse TOMT and TMC1 can directly interact in HEK 293 cells, and this interaction is modulated by His183 in TOMT. Thus, we propose a model of MET complex assembly where Tomt and the Tmcs interact within the secretory pathway to traffic Tmc proteins to the hair bundle.

## Introduction

Mechanoelectrical transduction (MET) is the process by which sensory hair cells convert mechanical force such as auditory and vestibular stimuli into electrical signals. The mechanosensitive organelle of the hair cell is the hair bundle, an apical collection of actin-filled stereocilia arranged in a staircase fashion. These stereocilia are tethered by interciliary links, including a *trans* heteromeric complex of Cadherin 23 (CDH23) - Protocadherin 15 (PCDH15) called the tip link (*Kazmierczak et al., 2007*). One potential location of the MET channel complex is at the lower end of the tip link where PCDH15 resides (*Beurg et al., 2009*). A commonly accepted model of hair-cell mechanotransduction postulates that excitatory deflections of the bundle towards the tallest stereocilia will tension the tip links, thereby transferring the mechanical force to the associated MET channel (*Basu et al., 2016*; *Corey and Hudspeth, 1983*; *Pickles et al., 1984*). How the MET channel complex is assembled to transduce mechanical stimuli is largely unknown.

Our current understanding is that the MET channel complex is composed of the tip link protein PCDH15, and the transmembrane proteins lipoma HMGIC fusion partner-like 5 (LHFPL5),

transmembrane inner ear (TMIE), and transmembrane channel-like proteins (TMC1 and TMC2) (*Beurg et al., 2015*; *Kawashima et al., 2011*; *Kurima et al., 2015*; *Maeda et al., 2014*; *Xiong et al., 2012*; *Zhao et al., 2014*). Although conclusive evidence is still lacking, the TMCs are currently the most promising candidates for the pore-forming subunit of the MET channel (*Corey and Holt, 2016*; *Wu and MullerMüller, 2016*). TMC proteins are present at the site of mechanotransduction at the stereocilia tips of hair bundles (*Beurg et al., 2015*; *Kurima et al., 2015*) and can interact directly with PCDH15 (*Beurg et al., 2015*; *Maeda et al., 2014*). In humans, mutations in *TMC1* are responsible for both recessive (DFNB7/11) and dominant (DFNA36) forms of nonsyndromic deafness (*Kurima et al., 2002*). In mice, *Tmc1/2* double knockouts have no conventional MET current (*Kawashima et al., 2011*). Several lines of evidence support the idea that TMC1/2 are the candidate pore-forming subunits of the MET channel. Hair cells expressing either *TMC1* or *TMC2* alone exhibit MET channel properties that are distinct from those observed when both proteins are present, suggesting that the TMCs may form heteromeric complexes (*Pan et al., 2013*). Consistent with this finding, the tonotopic gradient in MET channel conductance of outer hair cells (OHCs) is eliminated in *Tmc1* KO mice (*Beurg et al., 2014*). Moreover, the *Beethoven* (*Bth*) M412K amino acid change in mouse TMC1 reduces the calcium permeability and conductance of the MET channel (*Corns et al., 2016*; *Pan et al., 2013*). For these reasons, it is important to understand the role of TMC1/2 in mechanotransduction and how they form a functional unit with other members of the MET complex.

The zebrafish *mercury* (*mrc*) mutant was originally identified in a forward genetic screen for genes required for hearing and balance (*Nicolson et al., 1998*). Similar to those genes directly involved in hair-cell mechanotransduction (*cdh23/sputnik*, *pcdh15a/orbiter*), the *mercury* phenotype is characterized by (i) balance defects, (ii) an absence of the acoustic startle reflex, (iii) failure to inflate the swim bladder, (iv) lack of hair-cell-dependent calcium responses in the hindbrain, and (v) undetectable microphonic currents. Together, these phenotypes suggest that the *mercury* gene is essential for hair-cell mechanotransduction.

Here, we report that mutations in the zebrafish *transmembrane O-methyltransferase* (*tomt*) gene are causative for the *mercury* mutant phenotype. The Tomt protein is predicted to have a S-adenosylmethionine (SAM)-dependent methyltransferase domain that is most closely related to Catechol O-methyltransferase (Comt; EC 2.1.1.6). The human ortholog of the *tomt* gene is called Leucine Rich and O-Methyltransferase Containing (*LRTOMT*), a dual reading frame locus that codes for either Leucine-Rich Repeat Containing 51 (LRTOMT1 / LRRC51) or TOMT (LRTOMT2). Mutations in *LRTOMT2* are responsible for DFNB63, a non-syndromic, autosomal recessive form of human deafness that is characterized by severe to profound neurosensory hearing loss that can be congenital or prelingual (*Ahmed et al., 2008*; *Du et al., 2008*; *Kalay et al., 2007*; *Khan et al., 2007*; *Tlili et al., 2007*). No vestibular dysfunction has been described for DFNB63 patients. A mouse model of DFNB63 has also been reported. The *add* mouse (named for its attention deficit disorder-like symptoms) has a single R48L amino acid change in the *Tomt* (*Comt2*) gene, and behavioral analyses confirm that TOMT is required for both auditory and vestibular functions (*Du et al., 2008*). The major findings from the mouse model of DFNB63 were that TOMT exhibits modest O-methyltransferase activity toward the catecholamine norepinephrine, and that there is progressive degeneration of cochlear hair cells in TOMT-deficient mice. Based on these findings, the authors speculated that the hair-cell pathology was caused by deficient degradation of catecholamines. However, this hypothesis has not been tested.

Using the *mercury* mutant zebrafish as a model of DFNB63, we have found that Tomt-deficient hair cells have no mechantransduction current. Mechanotransduction in *mercury* mutants can be rescued by transgenic expression of either zebrafish Tomt or mouse TOMT, but not with the closely related Comt enzyme. This result suggests that catecholamine metabolism is not the cause of the MET defects. Instead, we show that Tomt is required for trafficking Tmc proteins to the hair bundle. We find that GFP-tagged Tmc1 and Tmc2b fail to localize to the hair bundle in *mercury* mutants, and that Tomt can rescue this trafficking defect. Furthermore, mouse TOMT and TMC1 can interact in HEK 293 cells, and this interaction is modulated by His183 in the putative active site of TOMT. Together, these data suggest that DFNB63 is unlikely to be a disease involving catecholamine metabolism. Rather, TOMT-deficient hair cells exhibit a specific defect in mechanotransduction that can be explained by a failure of TMCs to properly localize to the hair bundle. As such, we propose a

model where a TOMT-TMC interaction is required in the secretory pathway of hair cells for the proper integration of TMC proteins into the MET complex.

# Results

## Identification of the *mercury* mutation

The *mercury tk256c* locus (*Nicolson et al., 1998*) was initially mapped between the SSLP markers Z20009 (G41723) and Z858 (G40668) on the distal end of chromosome 15. Sequencing of known candidate genes within this region revealed no pathogenic mutations and mRNA in situ hybridization (ISH) for these genes did not reveal any transcripts with hair cell-enriched expression patterns (data not shown). To determine if there were any genes within the *mercury* critical region that were not annotated in the zebrafish genome assembly, we identified a region with conserved synteny on the stickleback (*Gasterosteus aculeatus*) groupI chromosome that contained many of the *mercury* candidate genes previously excluded by sequencing and ISH, including *inppl1a*, *stard10*, *clpb*, *phox2a*, and the folate receptor *IZUMO1R* (Assembly BROAD S1 - groupI:6160000–6236000). The stickleback ortholog of the human deafness gene *LRTOMT*/*DFNB63* was also present in this region (*tomt*, ENS-GACG00000007832). We used the stickleback Tomt protein sequence to identify *tomt*-postive contigs in the Sanger database of de novo zebrafish genome assemblies derived from Illumina sequencing of AB and TU double haploid individuals (http://www.sanger.ac.uk/cgi-bin/blast/submit-blast/d_rerio) (*Table 1*). Using this information, we cloned and sequenced the zebrafish *tomt* open-reading frame (ORF) from larval RNA (Accession number KX066099). Additionally, we amplified and sequenced each of the three coding exons and their flanking intronic regions from genomic DNA, and found that each *mercury* allele contains a nonsense mutation in the first exon of *tomt* (*Figure 1A*). These mutations truncate the protein product prior to (*tk256c*) or early within (*nl16*) the putative O-methyltransferase domain (*Figure 1B*) and are both predicted to be functional nulls.

## Zebrafish *tomt* is expressed exclusively in hair cells

To determine where the *tomt* gene is expressed, we performed whole mount mRNA ISH using the *tomt* coding sequence as a probe. At 28 hr post-fertilization (hpf), we observed ISH signal exclusively in the hair cells of the anterior and posterior maculae in the developing ear (*Figure 1C,D*). At 4 days post-fertilization (dpf), *tomt* is expressed specifically in hair cells of the inner ear and lateral line organ (*Figure 1E–G*). We found that the ISH signal is absent in *tomt*[nl16] mutants, suggesting that the G219A mutation causes nonsense-mediated mRNA decay (inner ear shown *Figure 1H,I*). This result

**Table 1.** Sanger AB and Tuebingen de novo genomic assembly contigs containing *tomt* coding sequence (GenBank: KX066099).

**AB strain (DHAB) Illumina de novo assembly**

| Contig Name | Exon | Region of *tomt* CDS |
| --- | --- | --- |
| Contig_000336392 | 1 | 1–262 |
| Contig_000381119 | 2 | 263–459 |
| Contig_000235950 | 3 | 460–780 |

Tuebingen strain (DHTu2) Illumina de novo assembly

| Contig Name | Exon | Region of *tomt* CDS |
| --- | --- | --- |
| c306000518.Contig1 | 1 | 1–60 |
| c279701478.Contig1 | 1 | 1–258 |
| c280900030.Contig1 | 1 | 141–262 |
| c301500577.Contig1 | 2 | 263–459 |
| c282600514.Contig1 | 3 | 730–780 |
| c282201256.Contig1 | 3 | 460–780 |
| c008000433.Contig1 | 3 | 460–599 |

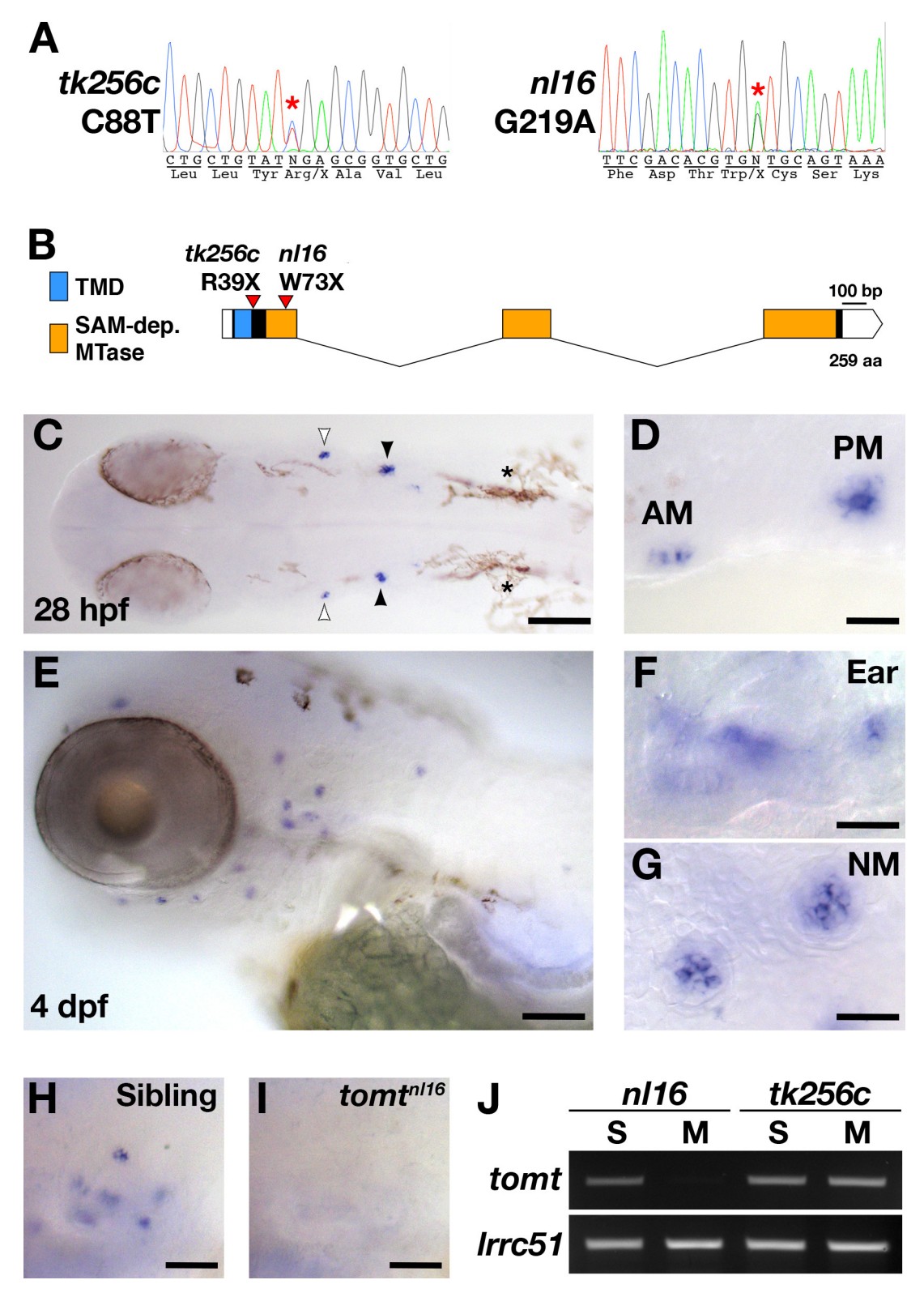

**Figure 1.** *mercury* mutations and *tomt* mRNA expression. (**A**) Representative chromatograms from heterozygous *mercury* mutants showing the C88T and G219A mutations for the *tk256c* and *nl16* alleles respectively. (**B**) Diagram of the predicted exon-intron structure for the *tomt* gene. Regions coding for the putative transmembrane domain (TMD, blue) and SAM-dependent O-methyltransferase domain (SAM-dep. MTase, orange) are shown, along with the positions of the *tomt^tk256c* R39X and *tomt^nl16* W73X mutations. (**C–G**) Whole mount mRNA in situ hybridization (ISH) for *tomt* in 28 hr post-

*Figure 1 continued on next page*

*Figure 1 continued*

fertilization (hpf) (C, D) and 4 days post-fertilization (dpf) (E-G) zebrafish larva. (C) At 28 hpf, *tomt* is expressed in exclusively in the presumptive anterior (AM) and posterior (PM) maculae of the developing ear, as indicated by the white and black arrow heads respectively. Pigment cells are indicated by asterisks (*). The embryo is shown in dorsal view with anterior to the left. (D) A close up of the AM and PM from the larva in C. (E–G) At 4 dpf, *tomt* is expressed exclusively in the hair cells of the inner ear (F) and lateral line neuromasts (G). Larva is shown in lateral view with anterior to the left and dorsal at the top. (H, I) ISH for *tomt* in a *tomt^{nl16}* WT sibling (H) and mutant (I) at 4 dpf. Inner ear sensory patches are shown. (J) RT-PCR for *tomt* and *lrrc51* from total RNA isolated from 5 dpf *tomt^{nl16}* and *tomt^{tk256c}* siblings (S) and mutants (M). Scale bars: 100 µm in C and E, 25 µm in D, F and G, 50 µm in F and G.

was confirmed using RT-PCR (*Figure 1J*). We were unable to amplify the *tomt* transcript from total RNA of homozygous *tomt^{nl16}* mutants but were still able to detect it in *tomt^{tk256c}* mutants. *lrrc51*, the gene that codes for the LRTOMT1 protein in humans, was used as a control.

## Tomt is enriched in the Golgi apparatus

*tomt* is predicted to code for a single-pass membrane protein featuring a short N-terminus followed by a transmembrane domain (TMD), with approximately 20 amino acids separating the TMD from the predicted O-methyltransferase catalytic domain. Immunolabel of TOMT in mouse cochlear hair cells localized the protein in the cytoplasm of inner and outer hair cells, and showed enrichment below the cuticular plate of OHCs (*Ahmed et al., 2008*). To determine the subcellular localization of Tomt in zebrafish hair cells, we used the hair-cell-specific promoter *myo6b* to mosaically express Tomt tagged with either GFP or an HA epitope at its C-terminus. Both Tomt-GFP and Tomt-HA are enriched in an apical membrane compartment above the nucleus (*Figure 2A,B*), similar to immunostain for mouse TOMT. Notably, no Tomt protein is detectable in the hair bundle.

In hair cells, the Golgi apparatus is positioned apical to the nucleus (*Sipe et al., 2013*). To confirm that the Tomt-enriched compartment is within the Golgi apparatus, we engineered a medial Golgi marker by fusing the first 110 amino acids of the zebrafish glycosyltransferase Mgat1a (*mannosyl (alpha-1,3-)-glycoprotein beta-1,2-N-acetylglucosaminyltransferase a*) to the mKate2 far-red fluorescent protein (Mgat1a_1-110-mKate2). This portion of Mgat1a includes the TMD and stem regions of the protein, and previous studies have shown that these regions are necessary and sufficient for localization and retention in the medial Golgi cisternae (*Tu and Banfield, 2010*). When co-expressed, Tomt-GFP and Mgat1a_1-110-mKate2 are partially co-localized in hair cells (*Figure 2C*). Compared to Mgat1a_1-110-mKate2, Tomt-GFP is more broadly distributed indicating that Tomt-GFP may be present at lower levels in the endoplasmic reticulum and the basolateral membrane in addition to the Golgi apparatus.

We noted that the organization of Tomt's predicted protein domains was reminiscent of Golgi-resident, Type II transmembrane glycosyltransferases like Mgat1 - a short N-terminus followed by a signal anchor TMD, and a stem region preceding the catalytic domain (*Tu and Banfield, 2010*). To test if the putative TMD and stem regions of Tomt are required for its localization, we expressed the first 45 amino acids of Tomt C-terminally tagged with GFP. Similar to Tomt-GFP and Tomt-HA, Tomt_1-45-GFP is also enriched in the Golgi apparatus (*Figure 2D*). Together, these results suggest that Tomt is a Golgi-enriched protein, and that the first 45 amino acids of Tomt are sufficient for its subcellular localization.

## Hearing loss is rescued by Tomt-GFP in *mercury* mutants

To confirm that mutations in *tomt* are responsible for the *mercury* phenotype, and show that the Golgi-enriched Tomt-GFP was functional, we asked whether the *myo6b:tomt-GFP* (*Tg(tomt)*) transgene could rescue the Acoustic Evoked Behavior Response (AEBR) in 6 dpf *tomt^{tk256c}* mutants (*Figure 3*). On average, wild-type, non-transgenic siblings responded to 72% of the acoustic stimuli ($n = 20$, 138/192 stimulations). In contrast, non-transgenic *mercury* mutants exhibited a startle response to 2% of stimuli, confirming that Tomt-deficient zebrafish are deaf ($n = 15$; 4/177 stimulations). Strikingly, we were able to restore full auditory function to *mercury* mutants with the *tomt-GFP* transgene. The AEBR of *Tg(tomt); tomt^{tk256}* larvae ($n = 15$; 118/158 stimulations) was statistically indistinguishable from wild-type, non-Tg and wild-type *Tg(tomt)* larvae ($n = 18$; 128/183 stimulations). These data confirm that *tomt* is the gene responsible for the *mercury* phenotype, and

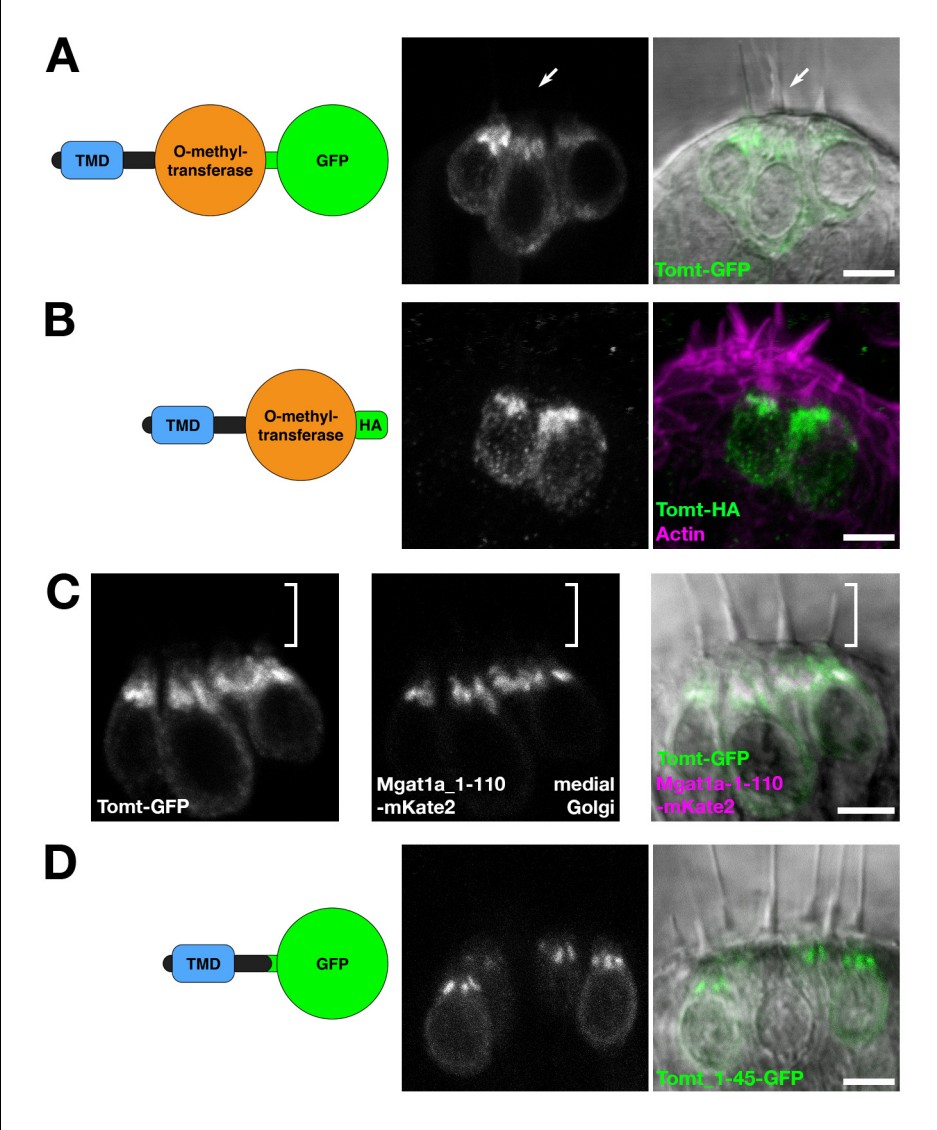

**Figure 2.** Tomt protein localization. (**A**) Diagram of the protein made from the *Tg(myo6b:tomt-GFP)* transgene plus an image of Tomt-GFP localization in live hair cells of the anterior crista in the inner ear at 3 dpf. The arrow points to an apical hair bundle. (**B**) Diagram of the protein made from the *Tg(myo6b:tomt-HA)* transgene plus immunolabel for the HA epitope showing Tomt-HA localization in the hair cells of the anterior crista in the inner ear at 4 dpf. Phalloidin stain (magenta) marks F-Actin. (**C**) Co-localization between Tomt-GFP and a medial Golgi marker (Mgat1a_1–110-mKate2) in live hair cells of the lateral crista (3 dpf). The hair bundle region is indicated by the white bracket. (**D**) Diagram of the protein made from the *Tg(myo6b:tomt_1–45-GFP)* transgene plus an image of Tomt_1–45-GFP localization in live hair cells of the lateral crista at 4 dpf. Scale bars: 5 µm in A-D.

indicate that the Golgi-enriched Tomt-GFP protein is fully functional and can rescue the behavioral phenotype of *mercury* mutants.

## Tomt is required for mechanotransduction in hair cells

The initial characterization of the *mercury* mutant suggested that the auditory and vestibular deficits were due to a lack of hair-cell mechanotransduction. The lateral line hair cells of *mercury* mutants lack microphonic currents and FM 1–43 dye uptake, both phenotypes associated with mechanotransduction defects (*Nicolson et al., 1998*; *Seiler and Nicolson, 1999*). To confirm whether Tomt-deficient hair cells have a specific defect in mechanotransduction, we performed electrophysiological

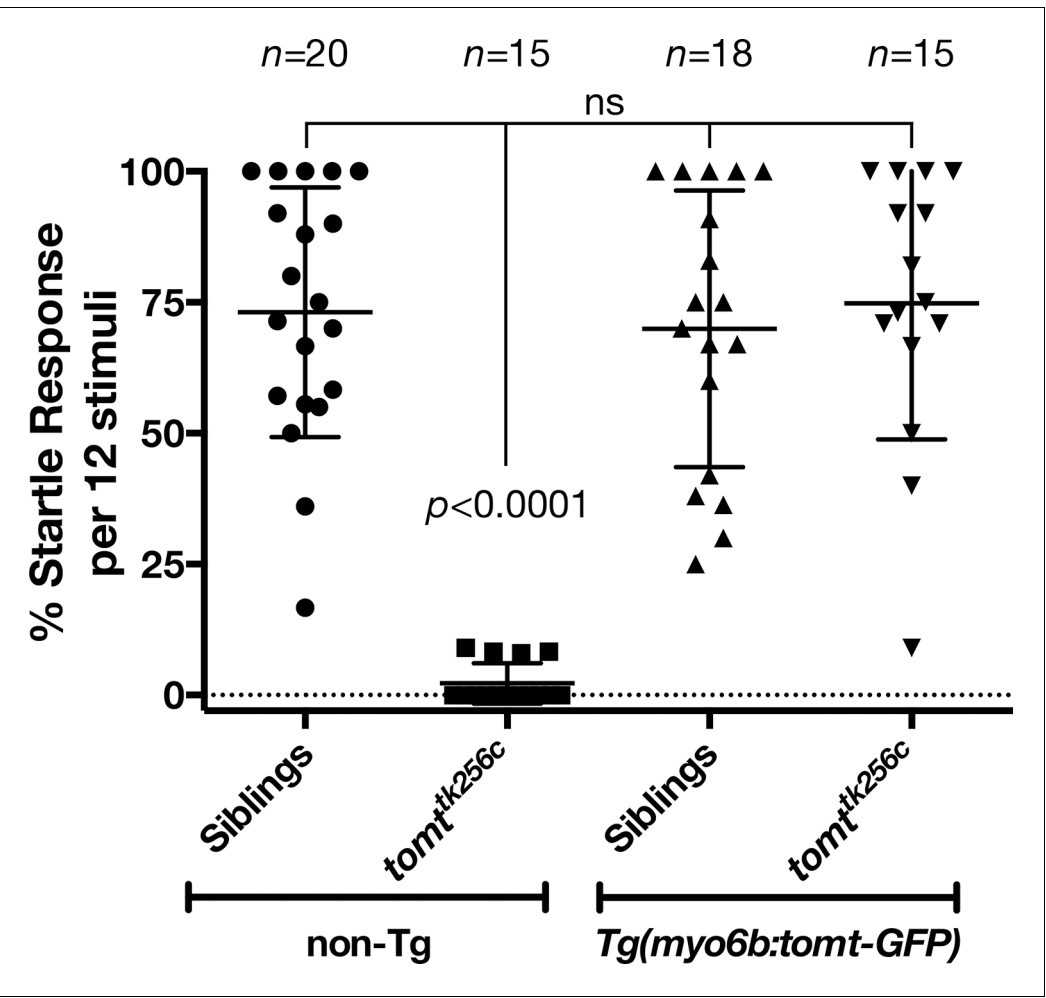

**Figure 3.** Auditory evoked behavior response (AEBR) in 6 dpf *tomt*[tk256c] siblings and mutants, with or without the *myo6b:tomt-GFP* transgene. The genotype of each group is labeled below, and the number of individuals analyzed for each genotype shown above. Each data point represents the percent of startle responses per trial of 12 stimuli for an individual larva. Error bars show the mean ± SD. Statistical comparisons were made by one-way ANOVA with Bonferroni's multiple comparison test.

recordings from lateral-line hair cells in wild-type and *mercury* mutants. Hair cells from the lateral line organ of wild-type and *tomt*[nl16] mutant zebrafish (3.0–5.2 dpf) showed a similar complement of $K^+$ currents (*Figure 4A,B*), in agreement with that previously described for wild-type hair cells (*Olt et al., 2016, 2014*). The size of the peak $K^+$ current measured at 0 mV was found to be similar between wild-type (261 ± 26 pA, $n = 4$) and mutant hair cells (352 ± 43 pA, $n = 3$) (*Figure 4C*). We then investigated whether the mechanoelectrical transducer (MET) current was affected in Tomt-deficient hair cells from 4.0 to 5.2 dpf zebrafish (*Figure 4D–F*). MET currents were elicited at the holding potential of –81 mV while displacing the hair bundles with sine wave stimuli from a piezo-electric-driven fluid jet (*Corns et al., 2016, 2014*). In wild-type hair cells, the size of the MET current was 86 ± 35 pA ($n = 4$ from four zebrafish, *Figure 4D,F*), with a resting open probability of the MET channel of 0.08 ± 0.03 ($n = 4$). By contrast, Tomt-deficient hair cells have no detectable MET current ($n = 10$ from six zebrafish; *Figure 4E,F*). The presence of the inward $Ca^{2+}$ current (inset in *Figure 4E*) was used to confirm hair cell-identity in *tomt*[nl16] mutants. The peak of the $Ca^{2+}$ current at –31 mV was 9.2 ± 2.4 pA ($n = 8$), which was similar to that previously reported (*Olt et al., 2016*).

We confirmed the absence of a functional MET channel in Tomt-deficient hair cells by using the styryl fluorescent dyes FM 1–43 and FM 4–64. These dyes are known to rapidly enter hair cells

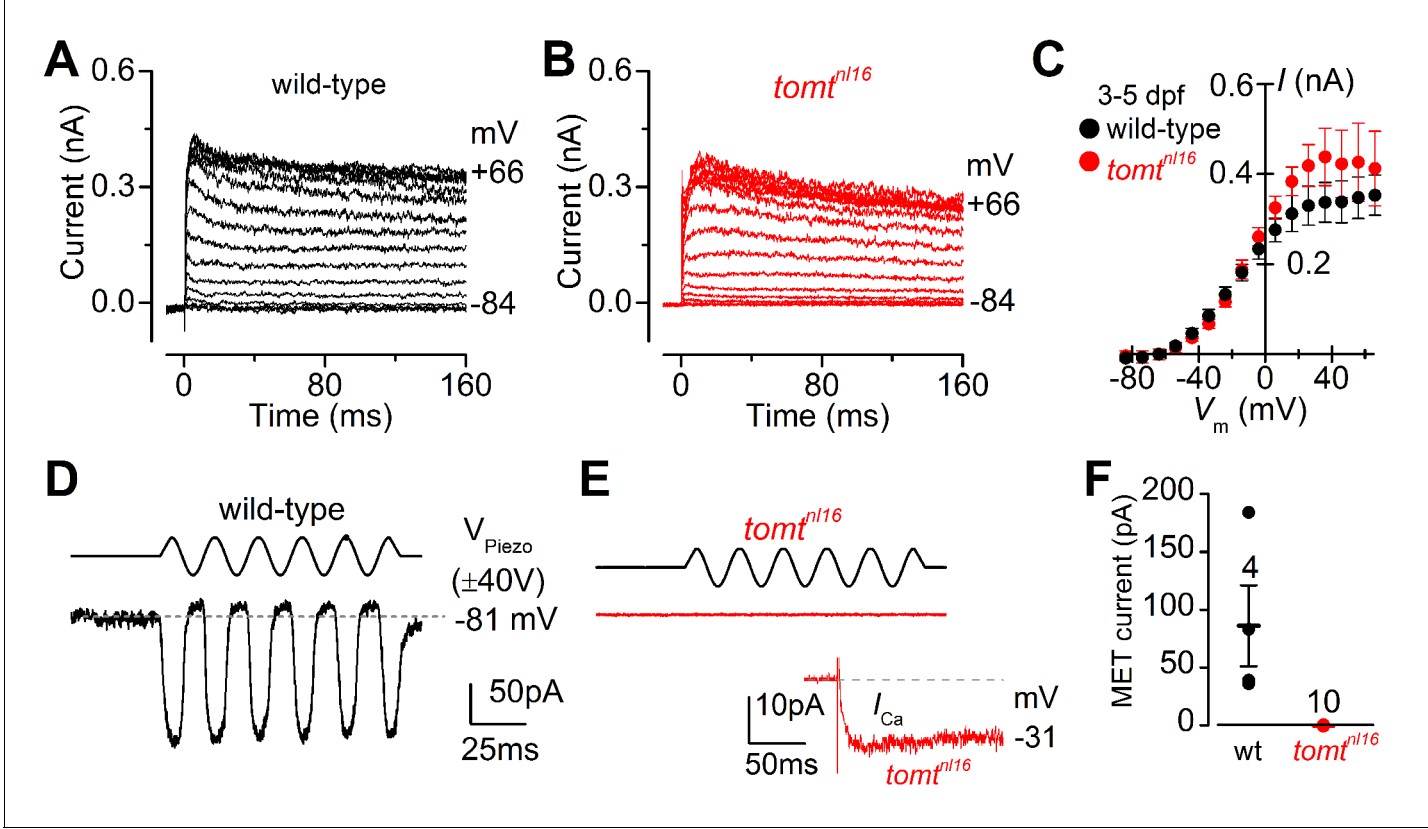

**Figure 4.** Tomt-deficient hair cells have no mechanotransduction (MET) current. (**A, B**) Examples of K+ currents recorded from lateral line hair cells in wild-type sibling (**A**) and *tomt*<sup>nl16</sup> mutant (**B**) zebrafish. Currents were elicited by depolarizing and hyperpolarizing voltage steps in 10 mV nominal increments from the holding potential of –84 mV. (**C**) Average peak current-voltage (*I-V*) curves from hair cells in wild-type (*n* = 4) and *tomt*<sup>nl16</sup> mutant (*n* = 3) hair cells, including those in panels A and B. (**D, E**) Saturating MET currents in 4 dpf zebrafish recorded from wild-type (**D**) and *tomt*<sup>nl16</sup> mutant (**E**) lateral line hair cells in response to a 50 Hz sinusoidal force stimulus to the hair bundles at the membrane potential of –81 mV, which is indicated by the dashed line (**D**). V<sub>Piezo</sub> indicates the driver voltage to the fluid jet, with positive deflections moving the hair bundles in the excitatory direction. Note the absence of the MET current in the *tomt*<sup>nl16</sup> mutant hair cell (**E**). The inset in panel E shows the calcium current recorded from the same cell in response to 150 ms depolarizing voltage steps in 10 mV increments from the holding potential of –81 mV. For clarity, only the peak Ca<sup>2+</sup> current trace at –31 mV is shown. (**F**) Average maximum MET current in both wild-type (wt) and mutant (*tomt*<sup>nl16</sup>) hair cells. Mean values in this Figure and text are quoted as means ± S.E.M.

through MET channels, thereby serving as a visual assay for basal channel activity (*Gale et al., 2000*; *Meyers et al., 2003*; *Nishikawa and Sasaki, 1996*; *Seiler and Nicolson, 1999*). Nascent hair cells of the lateral line organ will begin to label with FM dyes at 2 dpf (*Figure 5A*; *n* = 6 individuals, 2 NM each) (*Kindt et al., 2012*). However, Tomt-deficient hair cells did not label with FM 1–43 at this early developmental stage (*n* = 8 individuals, 2 NM each; *Figure 5A,B*), even though the neuromasts from *tomt*<sup>tk256c</sup> mutants contained the same number of hair cells (*Figure 5E*). To show that the lack of functional MET channels was not a case of developmental delay, we also quantified FM 1–43 uptake at 6 dpf, a stage when wild-type neuromasts contain an average of 17 hair cells per neuromast (*n* = 6 individuals, 2 NM each; *Figure 5C–E*). At 6 dpf, Tomt-deficient hair cells still did not label with FM 1–43 (*Figure 5C,D*). At this stage, we observed a significant decrease in the number of hair cells per neuromast in *mercury* mutants (*Figure 5E*, average of 13 HC / NM; *n* = 8 individuals, 2 NM each), consistent with what has been observed in other zebrafish mechanotransduction mutants (*Seiler et al., 2005*). The auditory and vestibular phenotypes of *mercury* mutants suggest that the hair cells of the inner ear also have defects in mechanotransduction. Injecting FM 1–43 into the ear of 6 dpf wild-type larvae led to robust labeling of inner ear hair cells (*Figure 5F*, top, lateral cristae shown, *n* = 5). However, like the lateral line organ, Tomt-deficient inner ear hair cells failed to label with FM 1–43 dye (*n* = 7). We did not observe any gross polarity (*Figure 5—figure supplement 1A,*

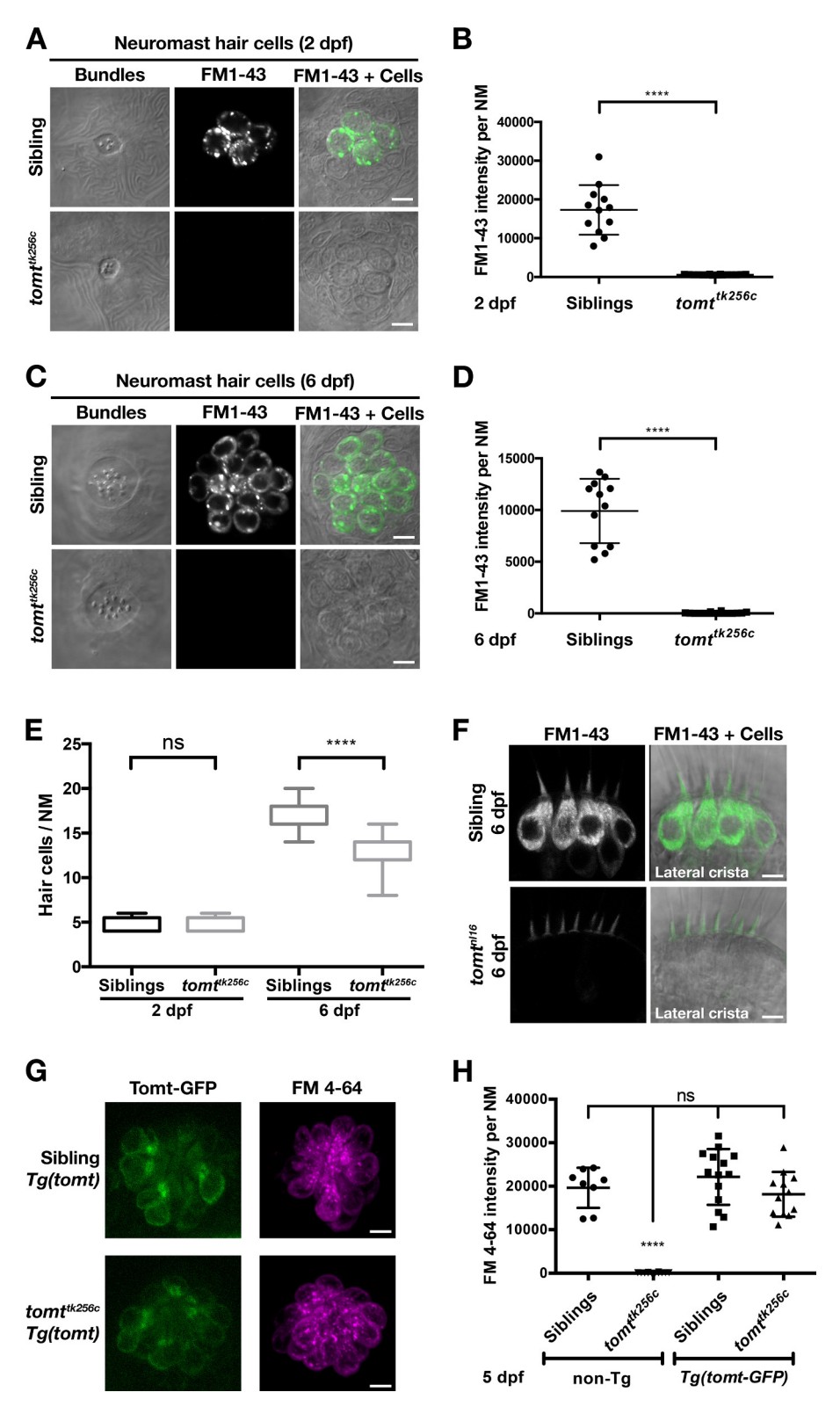

**Figure 5.** Tomt-deficient hair cells do not label with MET-channel permeant FM dyes. (A–D) FM 1–43 dye label of lateral line hair cells in 2 dpf (A, B) and 6 dpf (C, D) wild-type siblings and *tomt^{tk256c}* mutants. (A, C) Representative DIC images of NM hair bundles (left), fluorescence images of FM 1–43 in the same NMs (middle), and a merge of the FM 1–43 images with DIC images of the hair cell bodies (right) from 2 dpf (A) and 6 dpf (C) wild-type siblings and *tomt^{tk256c}* mutants. (B, D) Quantification of FM 1–43 fluorescence intensity per NM of 2 dpf (B) and 6 dpf (D) wild-type siblings (*n* = 6 larvae; *Figure 5 continued on next page*

*Figure 5 continued*
2 NMs each) and *tomt^tk256c* mutants (n = 8 larvae; 2 NMs each). Error bars are the mean ± SD. Asterisks indicate p<0.0001 by unpaired, two-tailed t-test. (E) Quantification of hair cell number per neuromast in 2 dpf and 6 dpf *tomt^tk256c* mutants and wild-type siblings (same as those shown in panels A – D). The box plots cover the 25th to 75th percentiles, and the whiskers represent the minimum and maximum counts. ns = not significant, asterisks indicate p<0.0001 by unpaired, two-tailed t-test. (F) Representative images of FM 1–43 labeling of inner ear hair cells in 6 dpf wild-type siblings (n = 5 larvae) and *tomt^nl16* mutants (n = 7 larvae). (G) Rescue of FM dye labeling in *mercury* mutants by stably expressed Tomt-GFP. Representative images of Tomt-GFP (left panels) and FM 4–64 (right panels) in lateral line NMs of a *Tg(myo6b:tomt-GFP)* wild-type sibling and a *Tg(myo6b:tomt-GFP);tomt^tk256c* mutants at 5 dpf. Tomt-GFP and FM 4–64 images are from the same NM for each genotype. (H) Quantification of FM 4–64 fluorescence intensity per NM for 5 dpf non-transgenic wild-type siblings (n = 6 larvae, 8 NMs), non-transgenic *tomt^tk256c* mutants (n = 6 larvae, 12 NMs), *Tg(myo6b:tomt-GFP)* wild-type siblings (n = 7 larvae, 14 NMs), and *Tg(myo6b:tomt-GFP);tomt^tk256c* mutants (n = 6 larvae, 12 NMs), including those NMs shown in F. Error bars are the mean ± SD. ns = not significant. Asterisks indicate p<0.0001 by one-way ANOVA with Bonferroni's multiple comparison test. Scale bars: 5 µm in A, C, F, and G.
The following figure supplements are available for figure 5:
**Figure supplement 1.** Hair bundle polarity and morphology in Tomt-deficient hair cells.
**Figure supplement 2.** Tomt function requires the putative transmembrane and enzymatic domains.

*B*) or morphological defects (*Figure 5—figure supplement 1C–E*) that could account for the lack of MET channel activity in the *mercury* mutant. Together with the electrophysiological recordings in *Figure 4*, these data demonstrate that Tomt-deficient hair cells lack functional MET channels, even during the initial development of mechanosensitivity.

## Tomt-GFP can restore MET channel activity to *mercury* mutants

Having established that we could rescue the deafness phenotype in *mercury* mutants with the *tomt-GFP* transgene (*Figure 3*), we then assayed for FM dye labeling in lateral line hair cells to determine whether Tomt-GFP could rescue the mechanotransduction defect. We observed that FM 4–64 label in wild-type *Tg(tomt)* neuromasts was statistically identical to their wild-type, non-transgenic counterparts (*Figure 5G,H*), indicating that extra Tomt protein does not appreciably alter the basal function of the MET channel. The lack of FM label was fully rescued in mutants stably expressing the *tomt-GFP* transgene specifically in hair cells (*Figure 5G,H*). We also observed full rescue of FM dye labeling in a transgenic line expressing Tomt-HA (*Tg(myo6b:tomt-HA-pA)*; *Figure 5—figure supplement 2A*). In contrast, we were unable to rescue FM 4–64 label using a cytoplasmic form of Tomt (HA-Tomt_45-259-GFP), modeled after human S-COMT (Accession # NP_009294) (*Figure 5—figure supplement 2B,C*), suggesting that the putative TMD is required for Tomt function. Conversely, the putative enzymatic portion of Tomt is also required for rescue, as Tomt_1-45-GFP had no effect on FM 4–64 label in wild-type or *mercury* mutants (*Figure 5—figure supplement 2D–F*). These data suggest that Tomt is necessary for mechanotransduction in sensory hair cells, and that both the transmembrane and enzymatic domains are required.

## Heat-shock inducible Tomt-GFP can restore MET channel activity to mature *mercury* mutant hair cells

The *myosin6b* promoter is active at all stages of zebrafish hair cell development (*Kindt et al., 2012*; *Seiler et al., 2004*). As such, rescue by the *myo6b:tomt-GFP* transgene does not address whether Tomt is actively required after hair cell maturation for normal MET channel activity. To supply Tomt protein to *mercury* mutant hair cells post-development, we used a heat-shock-inducible approach (*Figure 6A*). We chose a 5 dpf time point because the majority of neuromast hair cells are functionally mature by this time (*Kindt et al., 2012*), thereby allowing us to determine if transient expression of Tomt can restore MET to mutant hair cells that have developed without mechanotransduction.

Prior to heat shock treatment, no distinct Tomt-GFP signal was observed (*Figure 6B,C*), and little to no FM 4–64 hair cell label could be detected (*Figure 6B,D*). Post-heat shock, we observed a significant induction of Tomt-GFP and a significant increase in FM 4–64 intensity (p<0.0001; *Figure 6C, D*). These results demonstrate that an acute pulse of Tomt-GFP can restore MET channel activity to

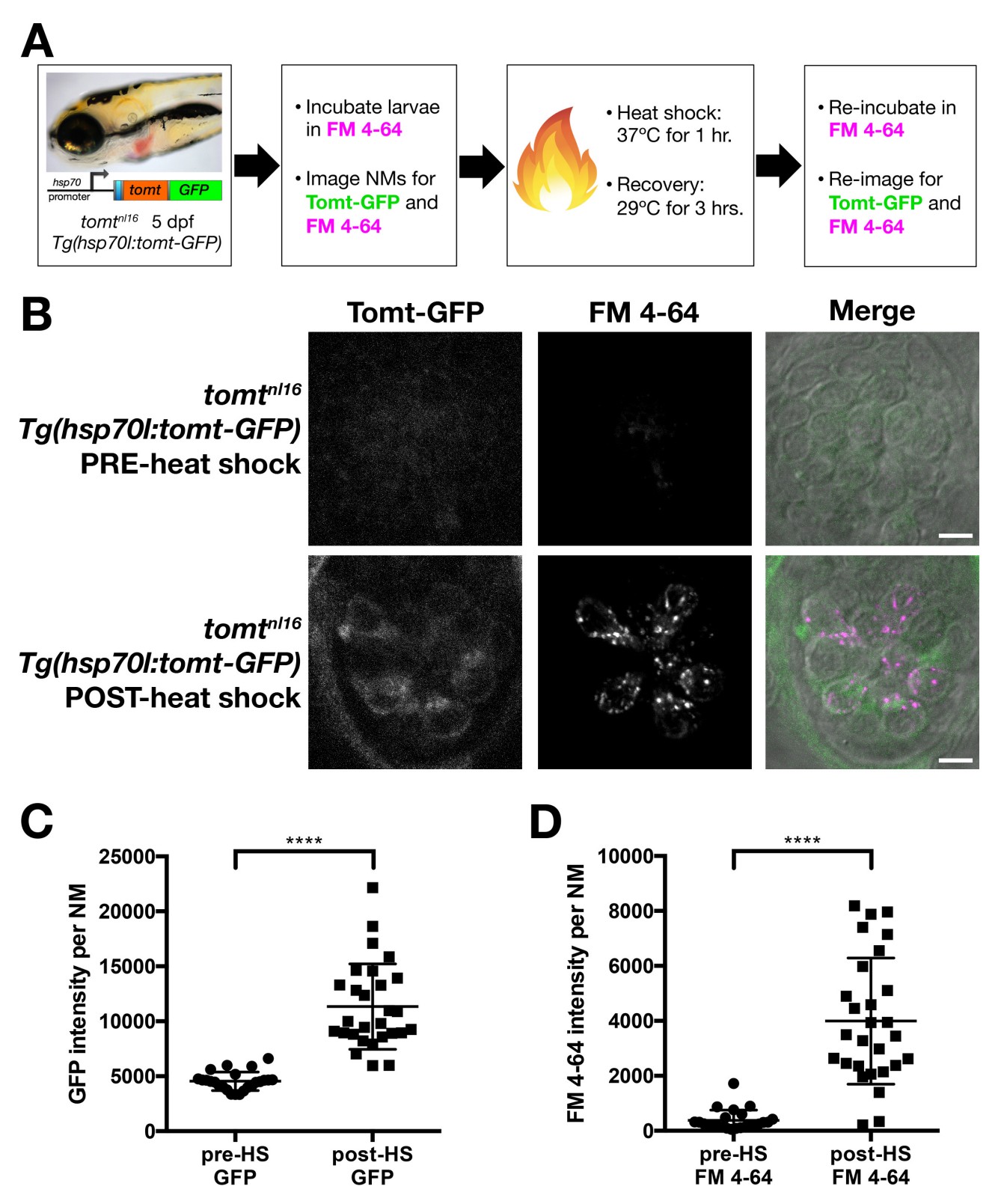

**Figure 6.** Restoration of MET channel activity in *mercury* mutant hair cells with heat shock-inducible Tomt-GFP. (**A**) Workflow for the heat-shock inducible *tomt-GFP* transgene experiment. (**B**) Representative images of Tomt-GFP (left panels), FM 4–64 (middle panels) and merged GFP, FM 4–64 and DIC channels (right panels) in lateral line NMs of 5 dpf *Tg(hsp70l:tomt-GFP)*; *tomt^{nl16}* larvae pre- and post-heat shock. (**C, D**) Quantification of GFP

*Figure 6 continued on next page*

*Figure 6 continued*

intensity (C) and FM 4–64 intensity (D) per NM of *Tg(hsp70l:tomt-GFP); tomt^{nl16}* larvae (*n* = 6) pre- and post-heat shock. Error bars are the mean ± SD. Asterisks indicate p<0.0001 by unpaired, two-tailed t-test. Scale bars: 5 μm in B.

previously silent hair cells, and can do so within 4 hr of initiating the heat-shock treatment. As such, these data suggest that Tomt plays an active role in MET channel function.

## Catechol-O-methyltransferase cannot rescue mechanotransduction channel activity in *mercury* mutants

Tomt is classed together with Catechol O-methyltransferaseCOMT in the EC 2.1.1.6 catechol O-methyltransferase protein family (*UniProt Consortium, 2015*). In their enzymatic domains, *Danio* Tomt (amino acids 43–259, Accession # ANO40802) is 44% identical and 68% similar to human S-COMT (Accession # NP_009294, amino acids 2–221) (*Figure 7A*). A previous study found that mouse TOMT exhibited some methyltransferase activity toward norepinephrine in vitro (*Du et al., 2008*). Based on these data, it was speculated that TOMT acts as a catechol O-methyltransferase in vivo, and that the deafness phenotype of the mouse mutant was caused by hair-cell degeneration resulting from a failure to properly metabolize catecholamines. If the *mercury* phenotype were caused by excess catecholamine, one would predict that increasing Comt activity would rescue mechanotransduction in Tomt-deficient hair cells. To test this hypothesis, we created a stable transgenic line expressing the zebrafish *comta* gene fused to GFP under the control of the hair-cell-specific *myo6b* promoter - *Tg(myo6b:comta-GFP)*. Homozygous *mercury* larvae expressing Comta-GFP exhibited auditory and vestibular defects identical to non-transgenic mutants (data not shown), and Comta-GFP had no effect on FM 4–64 label in Tomt-deficient hair cells (*Figure 7B,C*). These results suggest that deficient catecholamine metabolism in hair cells is not the cause of the *mercury* phenotype.

## COMT active site residues are not required for Tomt activity

The COMT enzyme catalyses the transfer of the methyl group from S-adenosylmethionine (SAM/AdoMet) to the meta-hydroxyl group (3-O-methylation) of its catechol substrate (*Axelrod and Tomchick, 1958*). The crystal structure for human COMT (structure PDB_3BWM) has revealed that the cluster of amino acids Asp141/191, His142/192, Trp143/193, and Lys144/194 are located in the active site (human S-COMT / MB-COMT amino acid numbering, *Figure 8A*) (*Rutherford et al., 2008*; *Vidgren et al., 1994*). Asp141/191 coordinates a requisite $Mg^{2+}$ ion, His142/192 and Trp143/193 interact with the SAM methyl donor, while Lys144/194 interacts with the catechol substrate and may aid in catalysis. This DHWK motif is conserved in all vertebrate COMT orthologs and some vertebrate TOMT proteins, most notably those from non-mammalian species. Interestingly, mammalian TOMT proteins retain only the histidine in this region (armadillo / mouse H183 and human H216, *Figure 8A*). The lack of conservation of these active site residues is surprising if TOMT shares the same substrates as COMT. To see if a mammalian TOMT was functional in zebrafish, we expressed mouse TOMT-GFP in *mercury* mutants (*Tg(myo6b:Mmu.Tomt-GFP)*). Using FM 4–64 label as an assay for MET channel activity in lateral line hair cells, we find that Mmu.TOMT-GFP can significantly restore mechanotransduction to *mercury* mutants (p<0.0001; *Figure 8B,C*), albeit not to wild-type levels (p<0.0001). This mild reduction in the efficacy of mouse TOMT to fully rescue FM label in *mercury* mutant zebrafish could be due to differences in TOMT localization or protein sequence relative to the endogenous zebrafish Tomt protein. However, as with the *Danio* Tomt-GFP transgene, homozygous *mercury* mutants expressing Mmu.TOMT-GFP are viable, fertile, and do not exhibit obvious behavioral phenotypes (data not shown).

The lack of amino acid sequence conservation between COMT and TOMT in the active site has been noted previously (*Ehler et al., 2014*). Although the native D182A substitution makes it unclear whether mammalian TOMT proteins can use $Mg^{2+}$, it has been suggested that H183 may serve as the catalytic residue due to the K185P substitution present in mammalian TOMT proteins (*Ehler et al., 2014*). To test whether H183 was required for TOMT function, we established a stable transgenic line of fish expressing *Mmu.Tomt_H183A-GFP* in hair cells. TOMT-H183A can significantly

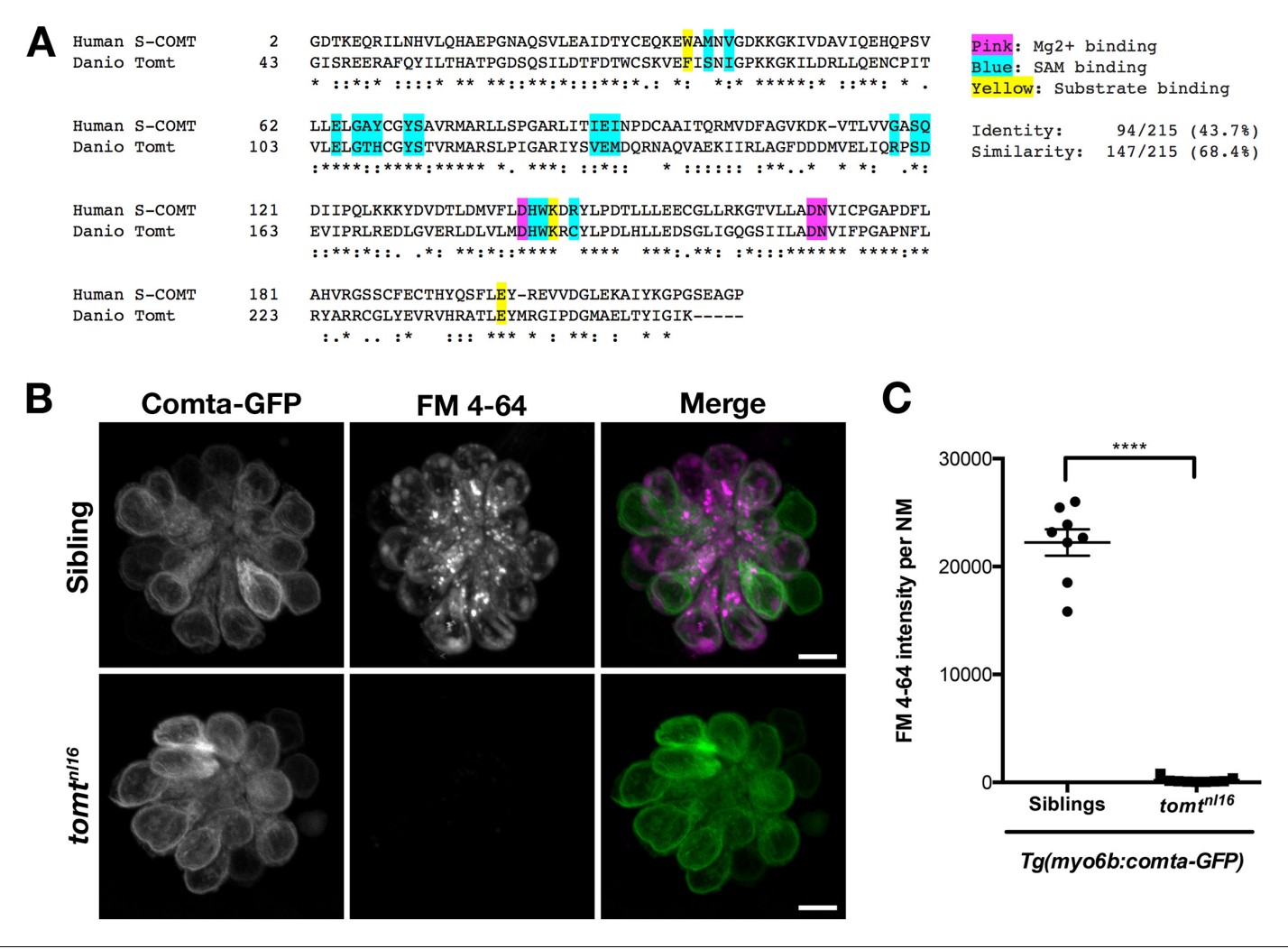

**Figure 7.** Comta-GFP cannot restore basal MET channel activity in *mercury* mutant hair cells. (**A**) Alignment between the putative enzymatic domains of human S-COMT (amino acids 2–221, Accession # NP_009294) and zebrafish Tomt (amino acids 43–259, Accession # ANO40802). Residues involved in $Mg^{2+}$-binding (pink), S-adenosylmethionine (SAM) binding (blue), and catechol substrate binding (yellow) are indicated, as determined by the crystal structure for human S-COMT (PDB_3BWM). Alignment legend - star (*) = conserved; colon (:) = conservative change; period (.) = semi-conservative change. (**B**) Representative images of Comta-GFP (left panels), FM 4–64 (middle panels) and merged GFP and FM 4–64 channels (right panels) in lateral line NMs of *Tg(myo6b:comta-GFP)* wild-type siblings and *tomt[nl16]* mutants at 6 dpf. (**C**) Quantification of FM 4–64 fluorescence intensity per NM for 6 dpf *Tg(myo6b:comta-GFP)* siblings (n = 5 larvae, 8 NMs) and *tomt[nl16]* mutants (n = 3 larvae, 9 NMs). Error bars are the mean ± SD. Asterisks indicate p<0.0001 by unpaired, two-tailed t-test. Scale bars = 5 μm in B, C.

rescue FM 4–64 label in *mercury* mutants at levels indistinguishable from wild-type mouse TOMT (p<0.0001; *Figure 8B,D*). And again, homozygous *mercury* mutants with the TOMT-H183A transgene are viable, fertile, and do not exhibit obvious behavioral phenotypes. These results indicate that none of these COMT active site residues are strictly required by TOMT to mediate mechanotransduction in hair cells.

## Localization of MET complex proteins Lhfpl5a, Pcdh15a, Tmie, Tmc1, and Tmc2b in *mercury* mutants

Two lines of evidence lead us to test whether Tomt regulates the trafficking and localization of MET complex components. First, Tomt itself is enriched in the Golgi apparatus and excluded from the hair bundle (*Figure 2*). Thus, it is well positioned within the secretory pathway to modulate protein trafficking or function. Secondly, since COMT activity cannot rescue the *mercury* phenotype, the

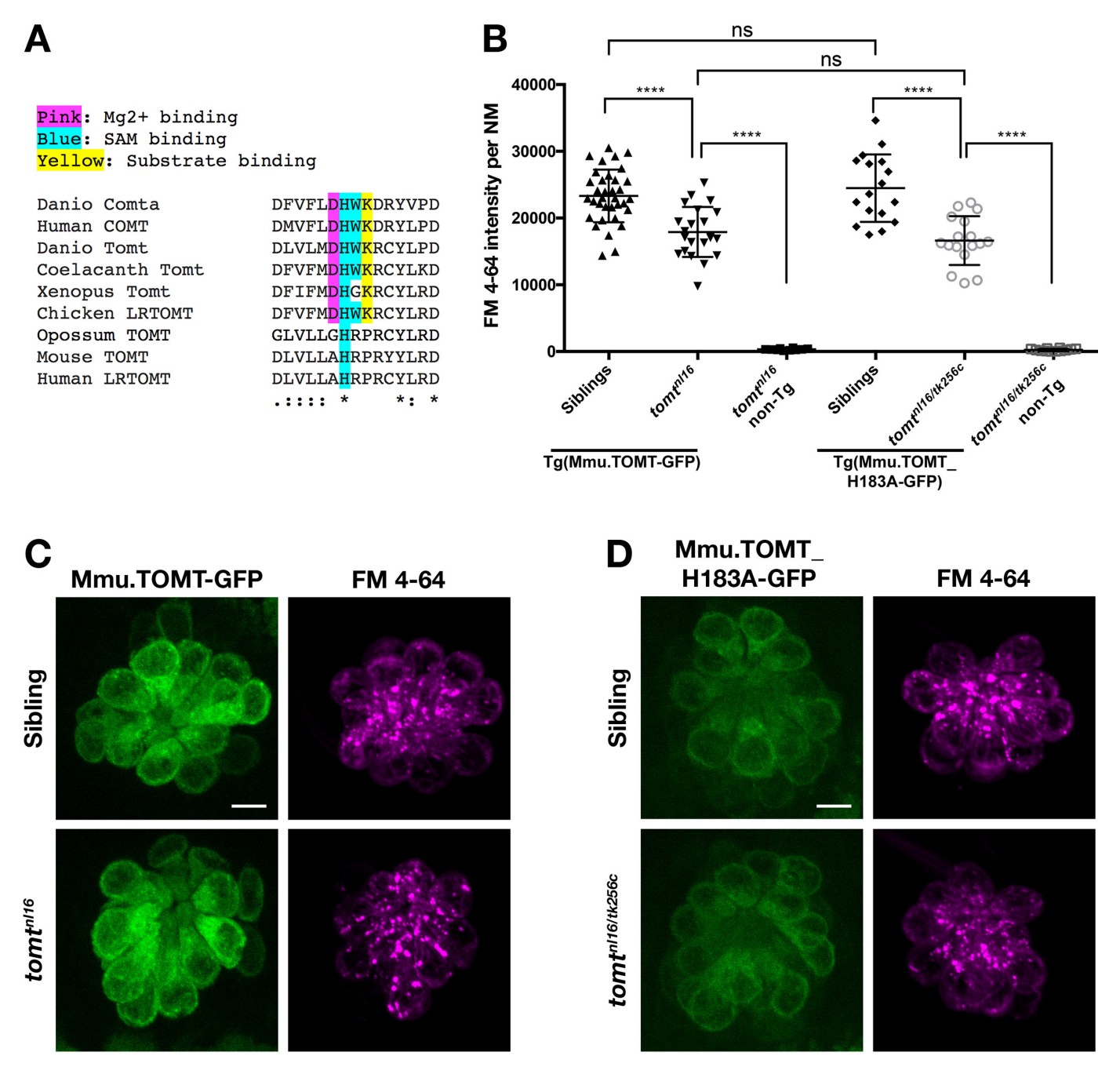

**Figure 8.** Mouse TOMT can restore basal MET channel activity to *mercury* mutant hair cells. (A) Alignment between putative active site residues in *Danio rerio* Comta (Asp178-Asp192; NP_001025328), Human MB-COMT (Asp186-Asp200; NP_000745), and Tomt/LRTOMT proteins from *Danio rerio* (Asp178-Asp19; ANO40802), Coelacanth (Asp178-Asp192; XP_006003643), *Xenopus tropicalis* (Asp206-Asp220; XP_004920324), Chicken (Asp178-Asp192; NP_001269010), Opossum (Gly177-Asp191; XP_016277512), Mouse (Asp177-Asp191; NP_001269017), and Human (Asp200-Asp214; NP_001138781. The shaded residues and alignment legend are the same as for *Figure 7*. (B) Quantification of FM 4–64 fluorescence intensity per NM for *tomt* mutants stably expressing either mouse TOMT-GFP (*myo6b:Mmu.Tomt-GFP; n = 8*, 22 NMs) or mouse TOMT-H183A-GFP (*myo6b:Mmu. Tomt_H183A-GFP; n = 7*, 17 NMs). For comparison, FM 4–64 fluorescence values are included for transgenic siblings (TOMT-GFP: *n = 12*, 35 NMs; TOMT-H183A: *n = 8*, 17 NMs) and non-transgenic mutants (*tomt^{nl16}: n = 4*; 10 NMs; *tomt^{nl16/tk256c}: n = 6*, 12 NMs). Error bars are the mean ± SD. Asterisks indicate p<0.0001 by one-way ANOVA with Bonferroni's multiple comparison test. (C) Representative images of Mmu.TOMT-GFP (left) and FM 4–64 (right) in lateral line NMs of a 5 dpf *Tg(myo6b:Mmu.Tomt-GFP)* sibling (top) and a *tomt^{nl16}* mutant (bottom). (D) Representative images of Mmu.TOMT_H183A-GFP (left) and FM 4–64 (right) in lateral line NMs of a 4 dpf *Tg(myo6b:Mmu.Tomt_H183A-GFP)* sibling (top) and a *tomt^{nl16/tk256c}* mutant (bottom). Images in C and D were near the mean of the FM 4–64 values shown in B. Scale bars = 5 μm in C, D.

mechanotransduction defect in *mercury* mutants is unlikely to be caused by a failure to metabolize catecholamines (*Figure 7*). Thus, we examined whether the MET complex proteins Lipoma HMGIC Fusion Partner-Like 5 (Lhfpl5a), Protocadherin 15a (Pcdh15a), Transmembrane Inner Ear (Tmie), and Transmembrane channel-like (Tmc) are correctly localized to the hair bundle of inner ear hair cells in *mercury* mutants (*Figure 9A–E*). For Pcdh15a, we used a previously characterized antibody that recognizes an N-terminal epitope present in both the CD1 and CD3 isoforms (*Maeda et al., 2017, 2014*). To localize Lhfpl5a, Tmie, Tmc1, and Tmc2b, we used stably integrated GFP or HA-tagged transgenes that are functional and able to rescue their respective mutant phenotypes (*Figure 9—figure supplement 1*; data not shown). Pcdh15a, GFP-Lhfpl5a, and Tmie-HA can still be trafficked to the hair bundle in *mercury* mutants (*Figure 9A–C*). However, neither Tmc1-GFP nor Tmc2b-GFP is detectable in the hair bundle of Tomt-deficient hair cells (*Figure 9D–E*), although the GFP signal remains in the cell body (*Figure 9F* – Tmc1-GFP; Tmc2b-GFP not shown). When overexpressed as

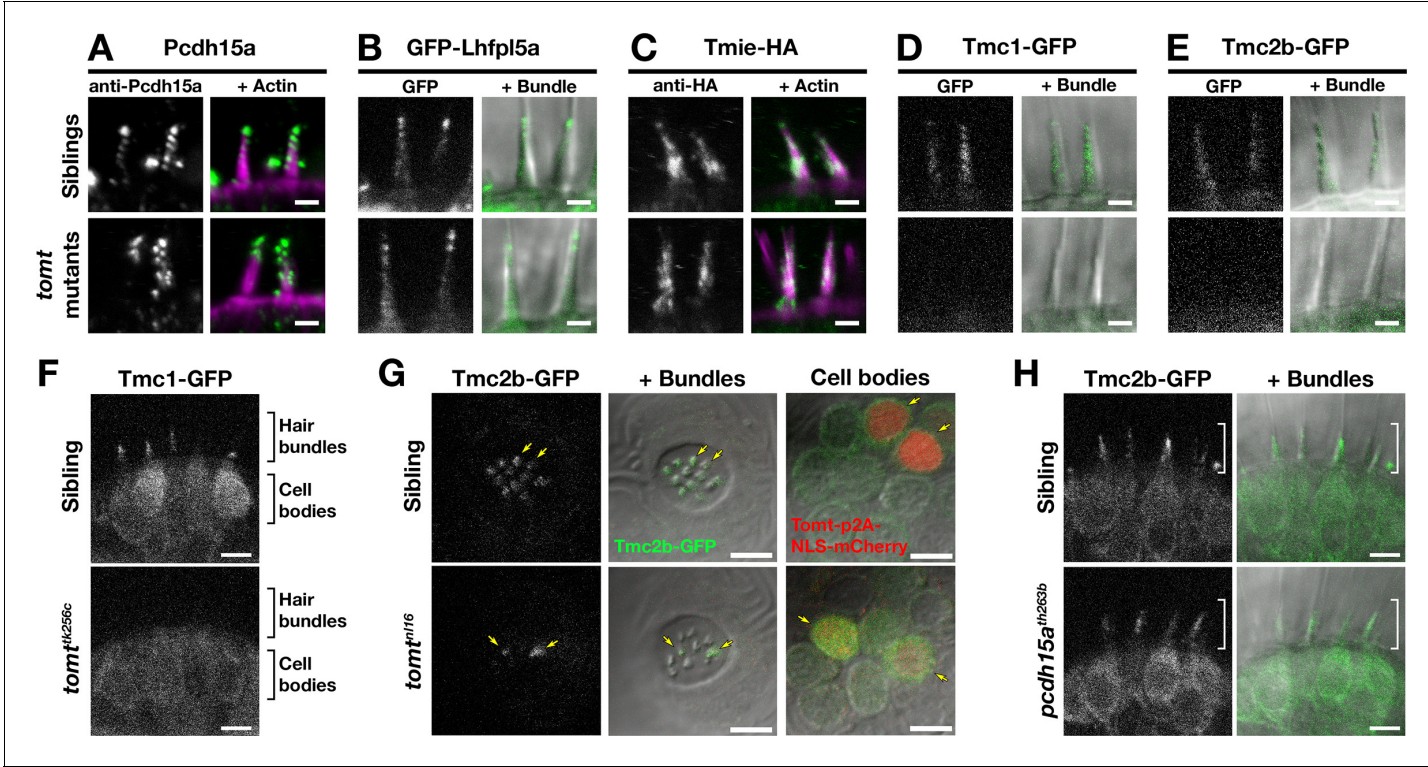

**Figure 9.** Hair bundle localization of MET complex proteins Lhfpl5a, Pcdh15a, Tmie, Tmc1, and Tmc2b in *mercury* mutants. (A–E) Representative images of (A) anti-Pcdh15a (Sibs and *tomt^{nl16}* n = 4 each), (B) GFP-Lhfpl5a (Sibs and *tomt^{tk256c}* n = 12 each), (C) Tmie-HA (Sibs and *tomt^{tk256c}* n = 15 each), (D) Tmc1-GFP (Sibs n = 17; *tomt^{tk256c}* n = 10), and (E) Tmc2b (Sibs n = 16; *tomt^{tk256c}* n = 12 in lateral cristae hair bundles at 4–5 dpf. Images in B, D, and E are from live larvae, while those in A and C are from immunolabeled, fixed specimens with phalloidin-labeled actin shown in magenta. (F) Optical sections through lateral cristae sensory patches showing the absence of Tmc1-GFP fluorescence specifically in the hair bundles of *tomt^{tk256c}* mutants, whereas GFP signal is present in the cells bodies. (G) Tomt can restore Tmc2b-GFP localization to the hair bundle of *mercury* mutant hair cells. Tmc2b-GFP fluorescence in neuromast hair cells of a 4 dpf *Tg(myo6b:tmc2b-GFP)* wild-type sibling (top) and *tomt^{nl16}* mutant (bottom) transiently expressing *tomt-p2a-nls-mCherry*. Note the presence of Tmc2b-GFP only in the Tomt-P2A-nls-mCherry expressing cells of the *tomt^{nl16}* mutant (yellow arrows, n = 3 individuals; 12 cells). (H) Representative images of Tmc2b-GFP (left panels) and merged GFP and DIC channels (right panels) in the lateral cristae of 4 dpf siblings (n = 6; top) and *pcdh15a^{th263b}* mutants (n = 6; bottom). White brackets indicated the hair bundle region of the hair cells. Scale bars = 2 μm in A-E, 5 μm in F-H.

The following figure supplements are available for figure 9:

**Figure supplement 1.** Tmc2b-GFP can restore basal MET channel activity to *tmc2b^{sa8817}* mutants.

**Figure supplement 2.** Transgenic expression of GFP-Lhfpl5a, Pcdh15aCD3-GFP, Tmc1-GFP, or Tmie-HA cannot restore basal MET channel activity to *mercury* mutant hair cells.

transgenes, none of these MET complex proteins can rescue basal MET channel activity in *mercury* mutants (*Figure 9—figure supplement 2*). Mosaic expression of a *tomt-P2A-NLS-mCherry* construct in *tomt*[nl16] mutants can rescue the bundle localization of Tmc2b-GFP (*Figure 9G*; *n* = 3 individuals, 12 cells), confirming that Tomt is required cell autonomously for Tmc trafficking to the hair bundle.

To determine if the defect in Tmc localization was secondary to a loss of mechanotransduction in Tomt-deficient hair cells, we imaged Tmc2b-GFP localization in *pcdh15a* (*orbiter*; *th263b*) mutants. This null allele (R360X) of *pcdh15a* exhibits a similar phenotype to *mercury* mutants: no microphonics, no acoustic startle response, and no FM dye label of lateral-line hair cells (*Maeda et al., 2017*; *Nicolson et al., 1998*). Tmc2b-GFP was still able to localize to the hair bundle of *pcdh15a* mutants, suggesting that Tmc protein localization is independent of Pcdh15a function and does not require mechanotransduction (*Figure 9H*). Together these results suggest that Tomt is specifically required for the correct trafficking of Tmc proteins to the hair bundle.

## Mouse TMC1 can directly interact with wild-type TOMT and TOMT-H183A

The observation that Tomt is required for Tmc trafficking to the hair bundle suggested that these proteins might interact. We tested this idea by co-expressing mouse TOMT and TMC1 in HEK 293 cells and performing co-immunoprecipitation experiments. TMC1-GFP was co-expressed with HA-tagged TOMT or TOMT-H183A, as well as the HA-tagged controls COMT, EZRIN (EZR), or RIα subunit of protein kinase A (PRKAR1A) (*Figure 10A,B*). HA immunoprecipitates were blotted for the presence of TMC1 (*Figure 10C,D*). TOMT and TMC1 can directly interact, and this interaction is reproducibly enhanced by the H183A change in TOMT (*Figure 10D*). There was no detectable interaction between TMC1 and COMT, EZR, or PRKAR1A. Likewise, the same pattern of interactions was detected when lysates were immunoprecipitated with anti-GFP and blotted for HA (*Figure 10E,F*). However, the TOMT-TMC1 interaction did not alter the subcellular localization of TMC1-GFP in HEK 293 cells; both TOMT and TMC1 were associated with intracellular membranes (*Figure 10—figure supplement 1*). These co-immunoprecipitation results suggest that TOMT and TMC1 can directly interact and support a model where TOMT interacts with the TMCs in the secretory pathway of hair cells to mediate TMC trafficking to the hair bundle.

## Discussion

In this study, we report that mutations in *transmembrane O-methyltransferase* (*tomt*) are responsible for the *mercury* mutant found in a screen for hearing and balance genes in zebrafish (*Nicolson et al., 1998*). *tomt* is the zebrafish ortholog of the human *LRTOMT2* gene, mutations in which are responsible for non-syndromic deafness DFNB63 (*Ahmed et al., 2008*; *Du et al., 2008*). Studies using the mouse model of DFNB63 suggested that TOMT functions as a catechol O-methyltransferase, and that the failure of hair cells to metabolize catecholamines leads to a degenerative phenotype and subsequent hearing loss (*Du et al., 2008*). However, progressive degeneration of hair cells is a common phenotype amongst mechanotransduction mutants in mice (*Alagramam et al., 2000*; *Kawashima et al., 2011*; *Longo-Guess et al., 2005*; *Mitchem et al., 2002*; *Steel and Bock, 1980*). It was not clear if aberrant catecholamine metabolism was truly responsible for the observed hair cell degeneration, nor whether Tomt-deficient hair cells had mechanotransduction defects prior to degenerating.

To clarify the role of Tomt in hair cell function, we used the zebrafish *mercury* mutant as a model for DFNB63. In the present study, we show that Tomt-deficient hair cells have a specific defect in mechanotransduction. Tomt-deficient hair cells have no evoked MET current and do not label with MET channel permeant FM dyes (*Figures 4* and *5*). The behavioral and physiological phenotypes can both be rescued by expression of Tomt-GFP specifically in hair cells (*Figures 3* and *5*). We determined that the absence of mechanotransduction was not due to a general developmental defect by using a heat-shock approach to express Tomt-GFP in mature hair cells. Heat-shock-inducible Tomt-GFP was able to restore MET channel function to mature mutant hair cells, indicating that Tomt-deficient hair cells are otherwise competent for mechanotransduction, but actively require Tomt function (*Figure 6*). Based on these results, we propose that a defect in hair cell mechanotransduction is the cause of hearing loss in *tomt* mutants and DFNB63 patients.

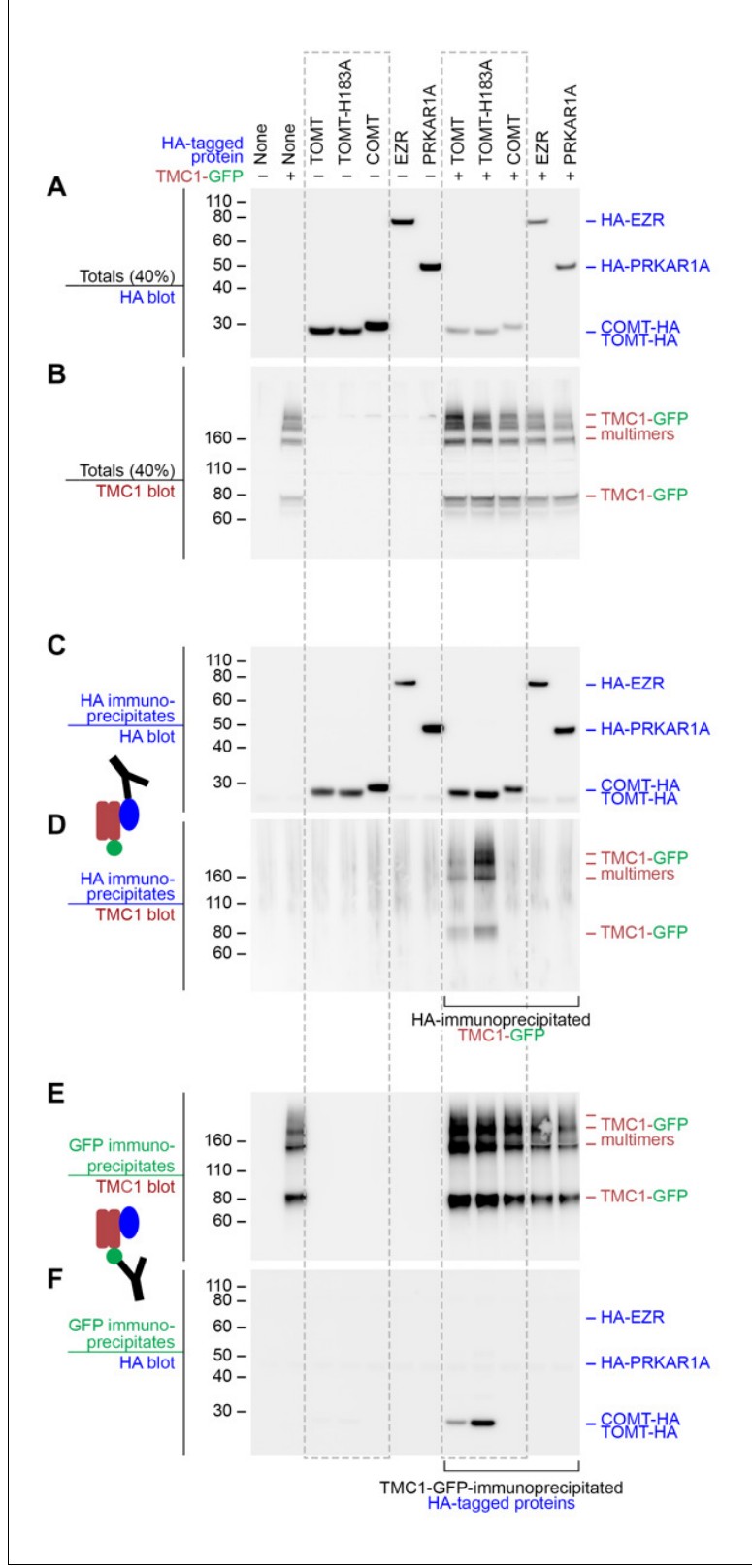

**Figure 10.** Mouse TOMT and TMC1 can interact in HEK 293 cells. Top labels: transfected proteins for all blots. Blue labels: HA-tagged proteins; Burgundy: TMC1; Green: GFP. (**A**) Anti-HA blot of totals (40% loaded relative to immunoprecipitates). (**B**) Anti-TMC1 blot of totals. (**C**) Anti-HA blot of HA immunoprecipitates. (**D**) TMC1 blot of HA immunoprecipitates. (**E**) TMC1 blot of GFP immunoprecipitates (immunoprecipitating GFP-TMC1). (**F**) HA blot

*Figure 10 continued on next page*

*Figure 10 continued*
of GFP immunoprecipitates. In both HA and GFP immunoprecipitation experiments, a robust interaction was detected between TOMT and TMC1; the H183A change in TOMT enhances this interaction. The HA-tagged controls COMT, EZR, and PRKAR1A did not interact with TMC1-GFP.
The following figure supplement is available for figure 10:

**Figure supplement 1.** TOMT-HA and TMC1-GFP in HEK 293 cells.

Our data suggest that the absence of mechanotransduction in *tomt* mutants is caused by defects in Tmc protein trafficking. Using GFP-tagged versions of zebrafish Tmc1 and Tmc2b, we show that Tmc proteins do not correctly localize to the hair bundle of Tomt-deficient hair cells (*Figure 9*). This trafficking defect can be rescued by transgenic expression of Tomt. Like Tomt-deficient hair cells, mouse TMC1/2 double knockout hair cells do not label with FM, have no evoked mechanotransduction currents, but have normal voltage-dependent currents (*Kawashima et al., 2011*). Likewise, *mercury* hair cells exhibit only mild defects in bundle morphology (*Figure 5—figure supplement 1*). This phenotype is similar to that observed in *Tmc1/2* double knockout mice at early stages when TMC-deficient hair cells have no mechanotransduction current but still have tip links and relatively normal bundle morphology. Furthermore, the time course of hair cell degeneration in TOMT-deficient mice is very similar to that observed for TMC-null mice (*Du et al., 2008*; *Kawashima et al., 2011*). Given the similarities between *Tomt* and *Tmc* mutant mice, it is likely that the zebrafish *mercury* mutant is equivalent to a triple *tmc1/2a/2b* knockout.

Typically, proteins that are involved in mechanotransduction are themselves localized in the hair bundle, the mechanosensitive organelle of the hair cell. However, neither Tomt-GFP, Tomt-HA, Tomt(1-45)-GFP nor Mmu.TOMT are detectable in the hair bundle. Rather, these proteins are localized to the secretory pathway, with the zebrafish proteins showing enriched localization in the Golgi apparatus (*Figures 2* and *8*). This intracellular location suggests that Tomt is regulating Tmc protein localization prior to the point at which the Tmcs are trafficked to the hair bundle. Although zebrafish Tomt is enriched in the Golgi compartment, this enrichment does not appear to be necessary for function. When transgenically expressed in zebrafish hair cells, Mmu.TOMT-GFP appears to be primarily located in the ER, yet can still restore MET channel activity to *mercury* mutants (*Figure 8*). We cannot exclude the possibility that Mmu.TOMT is also present in the Golgi apparatus, but enrichment within that organelle is not strictly required for function. Since other known members of the MET channel complex can localize to the bundle in *mercury* mutants (*Figure 9*), we propose that Tomt is actively required for the trafficking of Tmc proteins to the hair bundle where the Tmcs can then form a functional MET complex with Pcdh15, Lhfpl5, and Tmie. As such, LRTOMT / TOMT may be a suitable target for gene therapy, as has been shown for the TMCs (*Askew et al., 2015*). These data suggest that Tomt regulates Tmc protein trafficking (and therefore mechanotransduction) from within the secretory pathway, and does not directly participate in the MET complex.

The mechanisms by which Tomt regulates Tmc protein trafficking in hair cells are still not clear. However, we suggest that the mechanotransduction defects are not caused by a failure of hair cells to metabolize catecholamines. We find that Comta, a closely related O-methyltransferase, cannot restore mechanotransduction to *mercury* mutants (*Figure 7*). This result argues against the idea that Tomt is primarily responsible for catecholamine metabolism in hair cells. Additionally, transcriptomic and proteomic surveys of mouse and chick hair cells shows that *Comt* is endogenously expressed in hair cells and the surrounding cell types (*Scheffer et al., 2015*; *Shen et al., 2015*; *Shin et al., 2013*), yet is unable to compensate for the loss of TOMT. Furthermore, we show that COMT active site residues are not absolutely required for TOMT activity in hair cells (*Figure 8*), suggesting that TOMT and COMT proteins perform unique functions.

COMT can methylate a variety of catechol-containing substrates, with methylation of the 3'-hydroxyl (meta) favored over the 4'-hydroxyl (para) by about 5:1 in vitro (*Zhang and Klinman, 2011*). While an exhaustive survey has not been done, single amino acid substitutions in COMT can decrease its affinity for catechol substrates, decrease the meta:para ratio, and change the rate of catalysis (*Law et al., 2016*; *Zhang et al., 2015*; *Zhang and Klinman, 2011*). Although human COMT and Danio Tomt are 44% identical and 68% similar within their putative enzymatic domains

(*Figure 7*), those residues where they differ may have important consequences for Tomt methyltransferase activity toward catechols.

Using the COMT crystal structure as a guide, we find that there are some potentially important differences between the two enzymes, especially comparing mammalian COMT and TOMT. In S-COMT, the amino acid residues Asp141, Asp169, and Asn170 coordinate a $Mg^{2+}$ ion that is required to correctly orientate the hydroxyl groups of the catechol in the active site for methylation (*Vidgren et al., 1994*). Of these three residues, only TOMT Asp210 (orthologous to S-COMT Asp169) is conserved, suggesting that TOMT may not require a divalent ion in order to function. Mammalian TOMT proteins also differ from COMT with respect to other active site residues, most notably the putative catalytic residue Lys144 in S-COMT. Depending on the substrate, mutating COMT Lys144 to an alanine dramatically reduces or abolishes the methyltransfer reaction and can change the meta:para ratio (*Law et al., 2016*). It has been suggested that His183 could take over as the catalytic residue in TOMT due to the native Lys185Pro substitution (*Ehler et al., 2014*). However, the mouse TOMT-H183A-GFP protein can still rescue the mechanotransduction and behavioral defects in *mercury* mutants (*Figure 8*). Interestingly, the His183Ala change also enhances the biochemical interaction between TOMT and TMC1 in cultured cells (*Figure 10*). Together with the possibility that TOMT does not bind divalent cations, these results call into question whether TOMT functions as a catechol O-methyltransferase in vivo. Consistent with this idea is the finding that TOMT exhibited only modest catechol O-methyltransferase activity in vitro, even when supplied with supraphysiological levels of norepinephrine (*Du et al., 2008*). Thus, Tomt's *bona fide* physiological target has yet to be identified.

Given the evidence that TOMT is unlikely to be a functional catechol O-methyltransferase in vivo, the question now becomes: what is TOMT doing to regulate TMC protein trafficking in hair cells? Is it a methyltransferase and what is its substrate? Or is it performing a non-enzymatic function? There are precedents for methylation events regulating protein function and trafficking. Intriguingly, another COMT-related protein, Catechol O-methyltransferase domain containing protein 1 (COMTD1 / MT773), has been shown to stimulate epithelial Na+ channel (ENaC) currents (*Edinger et al., 2006*). Protein O-methylation is also involved in Ras protein trafficking (*Clarke, 1992*) and the function of a bacterial chemotaxis sensory system (*Falke et al., 1997*). However, the idea that TOMT is a protein methyltransferase is speculative at this point. Alternatively, the role of TOMT in sensory hair cells may be independent of an enzymatic function, as has been shown for some methyltransferases in other systems (*DebRoy et al., 2013*; *Dong et al., 2008*; *Perreault et al., 2009*). The protein-protein interaction between TOMT and TMC1 presents the possibility that TOMT acts as a chaperone to facilitate TMC protein folding or trafficking. However, we did not observe a redistribution of TMC1-GFP localization to the plasma membrane of HEK 293 cells when co-expressed with TOMT (*Figure 10—figure supplement 1*). This suggests that other factors in hair cells are involved in modulating TMC localization. More work is required to determine if TOMT is a methyltransferase in vivo, to identify its substrate, and to understand the functional consequences of the interaction between TOMT and the TMCs in sensory hair cells.

## Materials and methods

### Ethics statement

Zebrafish (*Danio rerio*) were maintained at 28°C and bred using standard conditions. Animal research complied with guidelines stipulated by the Institutional Animal Care and Use Committed at Oregon Health and Science University. Electrophysiological recordings from zebrafish larvae were licensed by the Home Office under the Animals (Scientific Procedures) Act 1986 and were approved by the University of Sheffield Ethical Review Committee. The following zebrafish mutant alleles were used for this study: *pcdh15a*[th263b], *tomt*[nl16], and *tomt*[tk256c] (*Nicolson et al., 1998*; *Seiler et al., 2005*). The *tmc2b*[sa8817] allele was obtained from the Wellcome Trust Sanger Institute Zebrafish Mutation Project (*Kettleborough et al., 2013*). All lines were maintained in a Tübingen or Tüpfel long fin wild-type background. For all experiments, we used larvae at 2–6 dpf, which are of indeterminate sex at this stage.

## Genotyping

Adult fish were genotyped by fin clipping; see *Supplementary file 1A* for *pcdh15a*[th263b], *tomt*[nl16/tk265c] and *tmc2b*[sa8817] genotyping primers. Mutant larvae were identified by either behaviour (auditory or vestibular defects) and/or lack of FM dye label of neuromasts. For those experiments where expression of a transgene rescued behaviour or FM dye label (*Figures 3*, *5* and *8*), homozygous mutant larvae were identified by single larvae DNA extraction (*Meeker et al., 2007*), followed by PCR and sequencing.

## RT-PCR, Gateway cloning, and Tol2 Gateway transgenesis

All primer sequences and expression constructs used in this study are provided in *Supplementary file 1*. RT-PCR for *tomt* and *lrrc51* was done by one-step RT-PCR (SuperScript III One-Step RT-PCR kit, Thermo Fisher Scientific, Waltham, MA) using 840 ng of total RNA from 5 dpf *tomt*[nl16] and *tomt*[tk256c] siblings and mutants following standard protocols. Gateway entry vector inserts were also cloned by one-step RT-PCR using gene specific primers with integrated Gateway recombination sites. Total RNA from 4 to 5 dpf larvae was used as the template for zebrafish genes, while mouse *Tomt* was cloned from WT mouse utricle total RNA. Gateway entry vectors were made by standard techniques (*Kwan et al., 2007*). The full-length ORF of *tmc1* and *tmc2b* were obtained by 5'-RACE or 3'-RACE by using total RNA extracted from whole larvae (SMARTer RACE cDNA Amplification Kit, Takara Bio, Mountain View, CA). *tmc1* and *tmc2b* ORFs were subcloned into the pDONR221 middle entry vector together with sequence coding for a peptide linker (GGGGS)x4 and a C-terminal monomeric EGFP tag. Construction of final Gateway expression vectors (*Supplementary file 1B*) and the generation of transgenic fish lines (*Supplementary file 1C*) were performed as previously described (*Kwan et al., 2007*). pcDNA3.1(+)Tomt-HA and pcDNA3.1(+) Comt-HA were made by standard cloning techniques using templates with NheI and XhoI sites added to the 5' end 3' ends by PCR. pcDNA3.1(+)Tomt-H183A-HA and pDONR221-*Mmu.Tomt_H183A* were made using the Quikchange Lightning site-directed mutagenesis kit (Agilent, Santa Clara, CA) according to the manufacturer's protocol.

## Acoustically evoked behavior response (AEBR)

Quantification of the larval AEBR was performed using the Zebrabox monitoring system (ViewPoint Life Sciences, Montreal, Canada) as previously described (*Einhorn et al., 2012*; *Maeda et al., 2017*). Each group of six larvae was subjected to two or three trials of 12 stimuli and, for each individual larva, the trial with best AEBR performance was used for quantification. Positive responses where spontaneous movement occurred in the second prior to the stimulus were excluded from analysis. Trials where spontaneous movement occurred for more than 6 of the 12 stimuli were also excluded from analysis.

## Electrophysiological recordings

For in vivo hair cell recordings, larvae (3.0–5.2 dpf) were briefly treated with MS-222 before being paralyzed by injecting 125 µM $\alpha$-bungarotoxin (Tocris, UK) into the heart (*Olt et al., 2014*). Whole-cell patch clamp experiments were performed at room temperature (21–24°C) from hair cells of the zebrafish primary neuromasts. Patch pipettes were made from soda glass capillaries (Harvard Apparatus Ltd, Edenbridge, UK) and had a typical resistance in the extracellular solution of 3–5 MΩ. In order to reduce the fast electrode capacitative transient, the shank of each capillary was coated with surfboard wax. Basolateral membrane current recordings were performed using the following intracellular solution: 131 mM KCl, 3 mM MgCl2, 1 mM EGTA-KOH, 5 mM Na2ATP, 5 mM Hepes-KOH, and 10 mM sodium phosphocreatine (pH 7.3). For mechanoelectrical transduction, the patch pipette was filled with an intracellular solution containing (in mM): 106 L-glutamic acid, 20 CsCl, 10 Na$_2$phosphocreatine, 3 MgCl$_2$, 1 EGTA-CsOH, 5 Na$_2$ATP, 5 HEPES and 0.3 GTP (the pH was adjusted to 7.3 with CsOH, 294 mOsmol/kg). Recordings were made with an Optopatch (Cairn Research Ltd, UK) or Multiplamp 900B (Molecular Devices, USA) amplifier. Data acquisition was performed using pClamp software with a Digidata 1440A data acquisition board (Molecular Devices, USA). Recordings were sampled at 5 kHz, low pass filtered at 2.5 kHz (8-pole Bessel) and stored on computer for offline analysis (Origin and PClamp). Membrane potentials in voltage clamp were corrected for the liquid junction potential, measured between electrode and bath solutions, of either −4 mV (KCl-based

intracellular) or–11 mV (L-glutamic acid-based intracellular). MET currents were elicited using a fluid jet from a pipette driven by a 25 mm diameter piezoelectric disc (*Corns et al., 2016*, *2014*; *Kros et al., 1992*). The fluid jet pipette tip had a diameter of 12–16 µm and was positioned at about 8–14 µm from the hair bundles in the neuromast. The distance of the pipette tip from the bundle was adjusted to elicit a maximal MET current. Mechanical stimuli were applied as steps or 50 Hz sinusoids (filtered at 1 kHz, 8-pole Bessel). Mean values are quoted in text and figures as means ± S.E.M.

## Immunostaining and whole mount mRNA in situ hybridization

Larvae were anesthetized with E3 plus 0.03% MS-222 and fixed in 4% paraformaldehyde/1x Phosphate Buffered Saline (PBS) for 4 hr at room temperature or overnight at 4°C followed by 5 × 5 min washes in PBS/0.1% Tween-20 (PBST). Fixed specimens were permeabilized with 0.5% triton-X in PBS (3 × 20 min), and blocked >2 hr in PBS/1% bovine serum albumin/1% DMSO/5% goat serum. Use of the anti-Pcdh15a antibody has been previously described (*Maeda et al., 2017*). To label HA-tagged Tomt or Tmie, larvae were incubated in a 1:1000 dilution of rat anti-HA clone 3F10 antibody (Sigma-Aldrich, St. Louis, MO) in block overnight at 4°C, washed 5 × 15 min in 1x PBS/0.01% Tween-20, incubated in a 1:1000 dilution of Dylight 549 goat anti-rat IgG (Jackson ImmunoResearch, West Grove, PA) with 1:1000 dilution of Alexa Fluor 488 phalloidin (Thermo Fisher Scientific), and washed again 5 × 15 min in PBS/0.01% Tween-20.

HEK 293 cells were plated in a 6-well cell culture dish and transfected using Effectene (Qiagen, Germantown, MD) following the manufacturer's protocol. Each well received either no plasmid, 0.4 µg of TMC1-GFP, or 0.4 µg TMC1-GFP and 0.4 µg TOMT-HA. After 20 hr, cells were rinsed briefly with PBS and fixed for 30 min in 4% formaldehyde at room temperature. Cells were rinsed 2x with PBS, then permeabilized and blocked for 1 hr in 0.2% saponin and 5% normal donkey serum. Cells were then incubated overnight with 1:500 anti-HA antibody (Proteintech, Rosemont, IL) diluted in blocking solution (5% normal donkey serum in PBS). Cells were rinsed 3x with PBS for 5–10 min each rinse and incubated for 3–4 hr with 1:1000 donkey anti-rabbit Alexa Fluor 568 secondary antibodies (Thermo Fisher Scientific) and 1:500 Alexa Fluor 647 phalloidin (Thermo Fisher Scientific). Cells were incubated with 1:5000 DAPI (Thermo Fisher Scientific) diluted in PBS for 10 min and the rinsed 3x with PBS for 5–10 min each rinse. Coverslips were then mounted on slides with Everbrite media (Biotium, Fremont, CA). Images were acquired using a 100 × 1.46 NA Plan-Apochromat objective on a Zeiss LSM780 with Airyscan processing.

Whole-mount mRNA in situ hybridization (ISH) and probe synthesis was performed essentially as described (*Erickson et al., 2010*; *Thisse and Thisse, 2008*). *tomt* antisense RNA probe synthesis was done using NotI linearized pCR4 plasmid containing the full length *tomt* coding sequence as a template. Specimens were mounted on a depression slide in 1.2% low-melting point agarose and imaged on a Leica DMLB microscope fitted with a Zeiss AxioCam MRc five camera using Zeiss Axio-Vision acquisition software (Version 4.5).

## FM dye labeling of hair cells

To label neuromast hair cells, groups of four larvae were incubated in a 3 µm solution of either FM 1–43 or FM 4–64 (Thermo Fisher Scientific) in E3 embryo media for 30 s, followed by three rinses in E3. To label hair cells of the inner ear, larvae were anesthetized with E3 plus 0.03% MS-222 and mounted laterally on a depression slide in 1.2% low-melting point agarose/E3. Approximately 2 nl of a 3 µm FM1-43 solution was injected directly into the otic capsule, and the larvae were immediately imaged. Because it is not possible to rinse out the FM dye from the otic capsule, some background staining of hair bundles is observed in *mercury* mutants.

## Imaging and quantification of fluorescence intensity

For imaging, live larvae were anesthetized with E3 plus 0.03% MS-222 and mounted laterally on a depression slide in 1.2% low-melting point agarose/E3 and imaged on a Zeiss LSM700 laser-scanning confocal microscope with a Plan Apochromat 40x/1.0 water lens and Zeiss Zen software. To quantify FM dye or GFP fluorescence intensity, unadjusted maximum projections were analyzed in Image J (v. 1.48). Fluorescence intensity is reported as the background-substracted Integrated Density value. Figures were assembled and adjusted for brightness and contrast in Adobe Photoshop

(CS6). Where relevant, individual channels were adjusted equally for siblings and mutants, and images chosen for Figures were near the mean of the group data. Because transient transgenesis can cause variation in expression levels between individual cells, the mCherry channel only in *Figure 9G* was differentially adjusted for brightness between the sibling and mutant images.

### Statistical analysis and replicates

For the purpose of this study, biological replicates are defined as the individual larvae analyzed in each experiment, the numbers of which are provided in the Figure legends. Data for quantification and statistical comparisons are taken from single experiments, although at least two technical replicates was performed for each experiment to confirm the results. All graphs and statistical comparisons were done using GraphPad Prism v.6.0h.

### Immunoprecipitation and immunoblotting

HEK 293 cells were seeded in multiwell plates with 10 cm wells at $1 \times 10^6$/dish. After 24 hr in culture, they were transfected with the indicated DNA combination using Effectene (Qiagen). To equalize protein expression, DNA was titrated to: 2 µg/dish TMC1-GFP, 0.25 µg/dish TOMT-HA, 0.5 µg/dish TOMT-H183A-HA, 0.1 µg/dish COMT-HA, 1 µg/dish HA-EZR, and 2 µg/dish HA-PRKAR1A. Total DNA was adjusted to 4 µg/dish using empty pcDNA3. After 48 hr, the medium was aspirated and the cells frozen rapidly at −80°C. Cell extracts were prepared using two 1 ml aliquots of RIPA buffer (50 mM Tris pH 8.0, 150 mM NaCl, 0.1% SDS, 1% NP-40, 0.5% deoxycholate) supplemented with protease inhibitors (Sigma-Aldrich, P8340). Insoluble material was removed by centrifuging at 90,700 x g ($r_{av}$).

Totals were prepared from 100 µl of extract with 100 µl 2X SDS-PAGE sample buffer (prepared from LifeTech LDS sample buffer together with DTT). Immunoprecipitations from 250 µl extract were accomplished with either 10 µl of 50 mg/ml Dynabeads MyOne Tosylactivated (#65502, Thermo Fisher Scientific) coupled with 2 mg/ml recombinantly produced anti-GFP (gift of Hongyu Zhao) for 1 hr at room-temperature, or 10 µl of anti-HA-agarose (clone HA-7; Sigma-Aldrich, #A2095) overnight at 4°C. Following incubation, beads were washed three times with RIPA buffer, and heated at 95°C for 10 min with two aliquots (90 µl) of 1X SDS-PAGE sample buffer (without DTT). After separation from the adsorbent, eluates were adjusted to 50 mM DTT. Totals were thus 40% of immunoprecipitates.

Samples were analyzed by SDS-PAGE using 4–12% gels with either MOPS (TMC1 immunoblots), or MES (HA immunoblots) running buffer (Thermo Fisher Scientific). Proteins were transferred to 0.45 µm PVDF membrane (Millipore, Billerica, MA), stained with India Ink, blocked with ECL PRIME blocking agent (GE Healthcare, Chicago, IL), and probed with rabbit anti-mmTMC1 (*Maeda et al., 2014*) or anti-HA (clone HA-7; Sigma-Aldrich) antibodies. Protein bands were visualized with HRP-coupled anti-rabbit or light-chain-specific anti-mouse (Jackson ImmunoResearch) and ECL PRIME (GE Healthcare) using a FujiFilm LAS3000 imaging system.

## Acknowledgements

This work was supported by the NIH (R01DC013572 and R01DC013531 to TN, R01HD072844 to AN, R01DC002368 and P30DC005983 to PGBG), the Max Planck Society (TN) and a Wellcome Trust grant 102892 (WM). Thank you to Daniel Gibson, Lisa Hayashi, and Leah Snyder for fish care, and to Matthew Avenarius (Barr-Gillespie lab) for mouse utricle RNA.

## Additional information

### Funding

| Funder | Grant reference number | Author |
| --- | --- | --- |
| National Institutes of Health | R01DC013572 | Teresa Nicolson |
| National Institutes of Health | NIH R01 DC013531 | Teresa Nicolson |
| Wellcome Trust | 102892 | Walter Marcotti |

| National Institutes of Health | R01DC002368 | Alex Nechiporuk<br>Peter G Barr-Gillespie |
| National Institutes of Health | P30DC005983 | Peter G Barr-Gillespie |

The funders had no role in study design, data collection and interpretation, or the decision to submit the work for publication.

## Author contributions

TE, Conceptualization, Data curation, Formal analysis, Validation, Investigation, Visualization, Methodology, Writing—original draft, Writing—review and editing; CPM, Data curation, Formal analysis, Validation, Investigation; JO, KH, EB-N, Data curation, Formal analysis, Investigation; RM, RC, Resources, Validation, Investigation; JFK, Data curation, Investigation, Visualization; AN, Resources, Supervision, Funding acquisition; PGB-G, Formal analysis, Supervision, Funding acquisition, Visualization, Writing—review and editing; WM, Data curation, Formal analysis, Supervision, Investigation, Writing—original draft, Writing—review and editing; TN, Conceptualization, Supervision, Funding acquisition, Investigation, Writing—review and editing

## Author ORCIDs

Timothy Erickson, http://orcid.org/0000-0002-0910-2535
Elisabeth Busch-Nentwich, http://orcid.org/0000-0001-6450-744X
Peter G Barr-Gillespie, http://orcid.org/0000-0002-9787-5860
Walter Marcotti, http://orcid.org/0000-0002-8770-7628
Teresa Nicolson, http://orcid.org/0000-0002-0873-1583

## Ethics

Animal experimentation: Animal research complied with guidelines stipulated by the Institutional Animal Care and Use Committed at Oregon Health and Science University (IP00000100). Electrophysiological recordings from zebrafish larvae were licensed by the Home Office under the Animals (Scientific Procedures) Act 1986 and were approved by the University of Sheffield Ethical Review Committee.

## Additional files

### Supplementary files

• Supplementary file 1. Primers (A), expression constructs (B) and transgenic fish lines (C) used in this study.

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
