## [Decision Letter]

Thank you for submitting your work entitled "Integration of Tmc1/2 into the mechanotransduction complex in zebrafish hair cells is regulated by Transmembrane O-methyltransferase" for consideration by *eLife*. Your article has been reviewed by 4 peer reviewers, and the evaluation has been overseen by a Reviewing Editor and a Senior Editor. The following individuals involved in review of your submission have agreed to reveal their identity: David Raible (Reviewer #2); Gregory I Frolenkov (Reviewer #3).

Our decision has been reached after consultation between the reviewers. Based on these discussions and the individual reviews below, we regret to inform you that your work will not be considered further for publication in *eLife*. Although this work examining the role of Tomt in zebrafish hair cells is interesting and potentially important, there are major concerns preventing fourth consideration at this time. The two biggest concerns of the Reviewers are the use of the resting potential as a proxy for mechanotransduction and the poor image quality for some of the figures. If the authors could address these major shortcomings, especially with new experiments that directly measure MET currents in the mutant, then a revision would be welcome.

*Reviewer #1:*

This is an interesting and potentially important paper examining the role in zebrafish hair cells of Tomt, a methyl-transferase protein previously shown to underlie human deafness. A large amount of data is presented on the distribution of the protein and the consequences of its mutation. The most significant conclusions are the absence of mechanotransduction in the Tomt mutant and its requirement for targeting of Tmc proteins to the hair bundle. Also important is the demonstration that Comta, a methyltransferase closely related to Tomt cannot restore transduction in *mercury* mutants, so eliminating an earlier hypothesis about the involvement of catecholamines. However, there are experimental weaknesses for some of the conclusions. Evidence used to argue for lack of MET currents in the mutant is a failure to incorporate FM1-43 and FM4-64 dyes (which normally permeate MET channels), and change in the resting potential in the mutant (which is interpreted as loss of resting MET current). Both of these arguments are indirect and the results could be explained by a shift in the position of the I-X curve rather than absence of MET channels. Although difficult, MET currents have been previously recorded in zebra fish hair cells (Ricci et al. 2013) and this must be attempted here to support the important conclusion. A few such records could be supplemented by measuring microphonics. The conclusion about the Tmc targeting suffers from poor images in Figure 10. In view of the significance of the conclusions, the supporting evidence must be strengthened. Poor imaging also undermines some of the other conclusions, such as in Figure 2 and Figure 9.

1) Figure 2. Tomt-GFP seems to be in the basolateral cytoplasm and/or membrane. Is this background? In the right hand image, where is the pink Golgi marker? Figure 2, these images are of limited value since the sub-cellular organization is unclear. An important conclusion is whether Tomt is present in the bundle (the authors state it is not) but given the low-resolution images this conclusion is questionable. It is too bad that sub-cellular localization is not demonstrated using immunogold labeling (of GFP) in electron-micrographs.

2) Figure 4. Use of the resting potential as a proxy for mechanotransduction is not convincing for several reasons. Firstly, the absolute values seem variable between measurements. Thus, Figure 4 shows controls around -50 mV with reduction in DHS to 60 mV in wild-type, whereas in Olt et al. (2014) controls are -60 mV with reduction in DHS to -67 mV. A manifestation of this is that the resting potentials in Figure 4 and Figure 4 in the presence of DHS are the same (-60 V) but different in Figure 4; thus in Figure 4, all three red points (Tomt mutants) should be at the same potential of -60 mV. However, a more important criticism is that with the mutant, the hyperpolarization might merely reflect a positive shift in the MET channel current-displacement relationship. A direct assay of transduction is needed; if this is not achievable, the figure should be removed because the K^+^ current data add nothing to the central argument. (Why is the resting MET current not evident in the I-V curves in Figure 4 – has it been subtracted as part of the leak?) Elsewhere in the paper, accumulation of FM dyes is used to indicate transduction even though that too has a similar drawback which is not commonly mentioned.

3) Figure 9. Lack of effect on bundle localization of Lhfpl5, Pchd15 and Tmie in the mutant is an important result but the examples shown in Figure 9 are ugly. Higher resolution views of the bundle might help, but there also seems significant background. The Tmie image looks especially bad, as though there many non-specific spots over the bundle. Better examples should be chosen. What is the significance of the statistically-significant differences in Figure 9?

4) Figure 10. Even worse problems exist here, especially in the Tmc1-GFP column (left-hand A, B, F, G). In none is the hair bundle labeling clear, and the cell body label is absent in B compared to A. In Figure 10, the Tomt mutant is non-zero, but there is no obvious label in B. The poor images should be compared to those in Figure 4 in Maeda et al. 2014. Perhaps images containing one hair-cell/bundle at higher resolution might be better.

5) Figure 12. This is a complicated figure. What is the significance of the TMC1 multimers in B, D and E? While the EZR, COMT and PRKAR1A negative controls are nice, they contribute to making the figure complicated and difficult to take in.

*Reviewer #2:*

LRTOMT2 is the gene responsible for the non-syndromic deafness DFNB63. The authors in this paper look at zebrafish with mutations in the zebrafish orthologue of this gene to uncover the role it plays in hearing. They show that zebrafish Tomt mutants show defects in mechanotransduction and that these defects are not due to developmental problems as they can be rescued by expressing Tomt in mature hair cells. It has previously been speculated that TOMT's role in hair cell activity was due to its O-methyltransferase activity and that deficient degradation of catecholamines lead to hearing loss. The authors provide evidence that this is not the case by showing that overexpression of the closely related Comt enzyme, which can also degrade catecholamines, does not rescue Tomt mutant phenotypes. Mutation of an amino acid necessary for methyltransferase activity does not affect the ability of Tomt to rescue mutants. The authors show that Tomt can directly interact with Tmc1 and Tmc2b, two components of the putative mechanotransduction channel, in 293 cells and that Tomt is required for the proper localization of these proteins to stereocilia.

While the authors demonstrate that zebrafish Comt cannot rescue Tomt mutants, this protein localizes to the ER instead of the Golgi. Mouse Tomt also appears to localize less strongly to Golgi. Would fusion of Tomt1-45, which is sufficient to target GFP to the Golgi, allow Comt to rescue?

In Figure 11D the two cells expressing tomt-P2A-nls-mCherry appear to show a significantly higher level of cytoplasmic Tmc2b in addition to localization of Tmc2b to stereocilia. It would be informative to see if there is consistently a change in cytoplasmic Tmc2 levels with changes in Tomt levels, as perhaps it is playing a role in protein stabilization as well as trafficking.

*Reviewer #3:*

This is an interesting manuscript describing unexpected role of the zebrafish transmembrane O-methyltransferase (Tomt) in the mechano-electrical transduction (MET) in the hair cells of the inner ear and lateral line neuromasts. It has been previously reported that mutations in human ortholog of the Tomt, LRTOMT, are responsible for DFNB63, a non-syndromic autosomal recessive deafness in humans. Mutations in Tomt gene in mouse also result in auditory and vestibular abnormalities. However, it was unknown how exactly all these mutations contribute to the auditory phenotype. This new study from Teresa Nicolson and collaborators as well as accompanying manuscript from Uli Muller group provide the first evidence for the involvement of Tomt in trafficking TMC1 and TMC2 (the putative components of the hair cell MET channels) to the stereocilia tips, thereby affecting hair cell mechanotransduction. Regulation of mechanotransduction by Tomt seems to be independent of its enzymatic activity and occurring not at stereocilia but within secretory pathway in the hair cell. Even though the mechanism of this regulation is unclear, this is the first identified molecule that may regulate trafficking of the MET channel components in the hair cells. Therefore, the study is significant despite few but essential drawbacks described below.

Based on the electrophysiological characterization presented in Figure 4, it is hard to agree with the authors' conclusion that the only effect of Tomt deficiency is "little to no resting MET current". If that would be the case, one would expect that the resting membrane potential in Tomt-deficient cells would be identical to the membrane potential of the wild type hair cells after blocking MET channels with dihydrostreptomycin (DHS), i.e. around -60 mV (Figure 4) – close to the equilibrium potential for K^+^. Instead, Tomt-deficient cells are constitutively depolarized to around -50 mV, exactly like the wild type hair cells with partially open MET channels (Fig, 4F). Independence of this constitutive depolarization from DHS may indicate that: (i) either Tomt deficiency causes insensitivity of still functioning MET channels to DHS, which is unlikely based on other data of the manuscript; or (ii) Tomt regulates other channels that are essential for establishing the resting membrane potential in hair cells. The latter possibility is particularly important and needs to be explored. Constitutive activation of some non-selective cation channels, for example, may depolarize the cell and change dramatically the intracellular Ca^2+^ levels affecting intracellular traffic of many molecules. The data presented in Figure 4 are not helpful in addressing this issue as they don't show the reversal potentials of the whole-cell currents.

Despite the fact that the *mercury* phenotype has been previously characterized, it would be helpful in some experiments of this study to assess MET function not only by MET channel-permeable fluorescent dyes but also by recording microphonic currents. For example, the experiments on Figure 8 or Figure 9—figure supplement 1 would greatly benefit from microphonic current recordings since the recovery of resting MET current may not correlate with the recovery of overall mechanosensitivity.

Besides the effects on TMC1 and TMC2, Tomt deficiency seems to affect also stereocilia localization of Pcdh15a and Tmie. In addition, electrophysiology data point to some abnormalities in Tomt-deficient hair cell besides MET current reduction (see above). Yet, the overall discussion and the take-home message of the paper are focused on TMC1 and TMC2 trafficking. This focus seems to be misleading. The data of the manuscript do not allow considering Tomt as a specific regulator of TMC1 and TMC2.

*Reviewer #4:*

The manuscript provides multiple lines of evidence favoring the authors' argument that transmembrane-O-methyltransferase serves as a chaperone that effects the translocation of the putative transduction-channel components Tmc1 and Tmc2 to their normal site atop the stereocilia of hair cells in the zebrafish. The work constitutes a demonstration of the mechanism of action of the mariner mutation in zebrafish. Moreover, inasmuch as it has been impossible heretofore to target the Tmcs to the membrane surface in heterologous cell systems, this is an interesting contribution that might facilitate the expression of the transduction complex.

Despite the positive aspects of the work, it suffers from a major flaw: although "mechanoelectrical transduction" is evoked again and again, that process is never tested! Instead, there is a bizarre proxy: modest changes in the current-evoked membrane potentials of mutant hair cells are interpreted as an indication that the resting mechanoelectrical-transduction current is absent. This is at best a weak and indirect indication of transduction, and the interpretation of the effect is highly questionable when one is dealing with a mutation whose phenotypic effects are uncertain.

Numerous investigators have demonstrated mechanoelectrical transduction in multiple experimental preparations, including hair cells of the zebrafish. Especially when the senior author is an accomplished electrophysiologist, there is no excuse for using ancillary effects of transduction and indirect arguments: the authors should in my estimation perform the necessary experiments.

---

## [Author Response]

*Reviewer #1:*

*This is an interesting and potentially important paper examining the role in zebrafish hair cells of Tomt, a methyl-transferase protein previously shown to underlie human deafness[…] In view of the significance of the conclusions, the supporting evidence must be strengthened. Poor imaging also undermines some of the other conclusions, such as in Figure 2 and Figure 9.*

*1) Figure 2. Tomt-GFP seems to be in the basolateral cytoplasm and/or membrane. Is this background? In the right hand image, where is the pink Golgi marker? Figure 2, these images are of limited value since the sub-cellular organization is unclear. An important conclusion is whether Tomt is present in the bundle (the authors state it is not) but given the low-resolution images this conclusion is questionable. It is too bad that sub-cellular localization is not demonstrated using immunogold labeling (of GFP) in electron-micrographs.*

We altered the text to clarify that Tomt-GFP and Tomt-HA may also be present in the endoplasmic reticulum or basolateral plasma membrane. Additionally, we altered and improved the clarity of Figure 2 by:

1] removing panel 2D (co-localization of Tomt-GFP and the medial Golgi marker in neuromast hair cells);

2] adding an arrow in Figure 2 to indicate the hair bundle, highlighting the observation that Tomt-GFP does not localize to the stereocilia;

3] adding brackets in Figure 2 to indicate the hair bundle region of the lateral crista, again highlighting the absence of Tomt-GFP signal in this region;

4] Figure 2, middle panel – added text "medial Golgi" to indicate the cisternae marked by the Mgat1a(1-110)-mKate2 protein.

*2) Figure 4. Use of the resting potential as a proxy for mechanotransduction is not convincing for several reasons. Firstly, the absolute values seem variable between measurements. Thus, Figure 4 shows controls around -50 mV with reduction in DHS to 60 mV in wild-type, whereas in Olt et al. (2014) controls are -60 mV with reduction in DHS to -67 mV. A manifestation of this is that the resting potentials in Figure 4 and Figure 4 in the presence of DHS are the same (-60 V) but different in Figure 4; thus in Figure 4, all three red points (Tomt mutants) should be at the same potential of -60 mV. However, a more important criticism is that with the mutant, the hyperpolarization might merely reflect a positive shift in the MET channel current-displacement relationship. A direct assay of transduction is needed; if this is not achievable, the figure should be removed because the K^+^ current data add nothing to the central argument. (Why is the resting MET current not evident in the I-V curves in Figure 4 – has it been subtracted as part of the leak?) Elsewhere in the paper, accumulation of FM dyes is used to indicate transduction even though that too has a similar drawback which is not commonly mentioned.*

We have included direct measurements of the MET current in wild type and *mercury* mutants. These measurements show that Tomt-deficient hair cells have no evoked MET current, consistent with the FM dye uptake experiments and the previously published microphonic recordings (Nicolson et al., 1998, Seiler and Nicolson, 1999).

*3) Figure 9. Lack of effect on bundle localization of Lhfpl5, Pchd15 and Tmie in the mutant is an important result but the examples shown in Figure 9 are ugly. Higher resolution views of the bundle might help, but there also seems significant background. The Tmie image looks especially bad, as though there many non-specific spots over the bundle. Better examples should be chosen. What is the significance of the statistically-significant differences in Figure 9?*

The original figure showed maximum projections of many bundles at low magnification. As requested, we have updated Figure 9 to show close up views of phalloidin-labeled hair bundles that are immunolabeled for Tmie and Pcdh15a, or close up views of live cells (GFP-tagged Tmc1/2 and Lhfpl5a). Additionally, we have removed the quantification of fluorescence intensity in the bundle to reflect that this assay is best interpreted qualitatively (i.e. can a protein traffic to the bundle in the absence of Tomt function or not).

*4) Figure 10. Even worse problems exist here, especially in the Tmc1-GFP column (left-hand A, B, F, G). In none is the hair bundle labeling clear, and the cell body label is absent in B compared to A. In Figure 10, the Tomt mutant is non-zero, but there is no obvious label in B. The poor images should be compared to those in Figure 4 in Maeda et al. 2014. Perhaps images containing one hair-cell/bundle at higher resolution might be better.*

We combined Figure 10 and Figure 9, and now show close up images of Tmc localization in the hair bundles as requested. We have re-imaged the Tmc1-GFP line and have included improved images of Tmc1 in the hair bundle (Figure 9). The GFP signal for both Tmc1 and Tmc2b is concentrated at the beveled edge of the hair bundles (see DIC overlays). We disagree that there isn’t a signal in the cell bodies of Tmc1-GFP expressing hair cells and have provided separate panels clearly indicating the Tmc1-GFP signal in the somas of mutant hair cells. Figure 4 in Maeda et al. (2014) shows high-level, transient expression of a short 117 aa N-terminal fragment of Tmc2a with a CAAX prenylation signal in zebrafish hair cells. The prenylation site enhances the localization of the GFP signal to the plasma membrane within the hair bundle. The Tmc2a fragment was used to show a dominant negative effect on mechanosensitivity. The images of this fragment are not comparable to images showing expression of full-length, stable transgenes of Tmc1 or Tmc2b in zebrafish hair cells.

*5) Figure 12. This is a complicated figure. What is the significance of the TMC1 multimers in B, D and E? While the EZR, COMT and PRKAR1A negative controls are nice, they contribute to making the figure complicated and difficult to take in.*

We have added some color labels and diagrams to make this figure easier to interpret.

*Reviewer #2:*

*[…] While the authors demonstrate that zebrafish Comt cannot rescue Tomt mutants, this protein localizes to the ER instead of the Golgi. Mouse Tomt also appears to localize less strongly to Golgi. Would fusion of Tomt1-45, which is sufficient to target GFP to the Golgi, allow Comt to rescue?*

This is a very interesting line of experiments that will be included in a more thorough structure-function analysis of Tomt. However, we feel they are outside the scope of the current study.

*In Figure 11D the two cells expressing tomt-P2A-nls-mCherry appear to show a significantly higher level of cytoplasmic Tmc2b in addition to localization of Tmc2b to stereocilia. It would be informative to see if there is consistently a change in cytoplasmic Tmc2 levels with changes in Tomt levels, as perhaps it is playing a role in protein stabilization as well as trafficking.*

Currently, we cannot say whether Tomt is required for Tmc stability and/or trafficking, only that the Tmcs do not make it to the hair bundle without Tomt function. Regardless, we did not consistently observe an increase in Tmc protein levels when Tomt was overexpressed.

*Reviewer #3:*

*[…] Based on the electrophysiological characterization presented in Figure 4, it is hard to agree with the authors' conclusion that the only effect of Tomt deficiency is "little to no resting MET current". If that would be the case, one would expect that the resting membrane potential in Tomt-deficient cells would be identical to the membrane potential of the wild type hair cells after blocking MET channels with dihydrostreptomycin (DHS), i.e. around -60 mV (Figure 4) – close to the equilibrium potential for K^+^. Instead, Tomt-deficient cells are constitutively depolarized to around -50 mV, exactly like the wild type hair cells with partially open MET channels (Fig, 4F). Independence of this constitutive depolarization from DHS may indicate that: (i) either Tomt deficiency causes insensitivity of still functioning MET channels to DHS, which is unlikely based on other data of the manuscript; or (ii) Tomt regulates other channels that are essential for establishing the resting membrane potential in hair cells. The latter possibility is particularly important and needs to be explored. Constitutive activation of some non-selective cation channels, for example, may depolarize the cell and change dramatically the intracellular Ca^2+^ levels affecting intracellular traffic of many molecules. The data presented in Figure 4 are not helpful in addressing this issue as they don't show the reversal potentials of the whole-cell currents.*

*Despite the fact that the mercury phenotype has been previously characterized, it would be helpful in some experiments of this study to assess MET function not only by MET channel-permeable fluorescent dyes but also by recording microphonic currents. For example, the experiments on Figure 8 or Figure 9—figure supplement 1 would greatly benefit from microphonic current recordings since the recovery of resting MET current may not correlate with the recovery of overall mechanosensitivity.*

We have included direct measurements of the MET current in wild type and *mercury* mutants as requested. These measurements show that Tomt-deficient hair cells have no evoked MET current, consistent with the results of the FM dye uptake experiments and the previously published microphonic recordings. We observe excellent correlation between behavior, FM dye uptake and microphonics in this and other studies from our lab. While the FM dye assay does not directly quantify hair cell mechanosensitivity, we feel that it is sufficient to demonstrate whether a particular treatment or protein can restore MET channel function, at least in its resting state.

*Besides the effects on TMC1 and TMC2, Tomt deficiency seems to affect also stereocilia localization of Pcdh15a and Tmie. In addition, electrophysiology data point to some abnormalities in Tomt-deficient hair cell besides MET current reduction (see above). Yet, the overall discussion and the take-home message of the paper are focused on TMC1 and TMC2 trafficking. This focus seems to be misleading. The data of the manuscript do not allow considering Tomt as a specific regulator of TMC1 and TMC2.*

With respect to the localization of MET complex proteins, we have removed the quantification of fluorescence intensity in the bundle to reflect that this assay is best interpreted qualitatively (i.e. can a protein traffic to the bundle in the absence of Tomt function or not). The focus is on the Tmcs because their trafficking is most affected in the absence of Tomt function, whereas the localization of other MET components is not impaired.

*Reviewer #4:*

*[…] Despite the positive aspects of the work, it suffers from a major flaw: although "mechanoelectrical transduction" is evoked again and again, that process is never tested! Instead, there is a bizarre proxy: modest changes in the current-evoked membrane potentials of mutant hair cells are interpreted as an indication that the resting mechanoelectrical-transduction current is absent. This is at best a weak and indirect indication of transduction, and the interpretation of the effect is highly questionable when one is dealing with a mutation whose phenotypic effects are uncertain.*

*Numerous investigators have demonstrated mechanoelectrical transduction in multiple experimental preparations, including hair cells of the zebrafish. Especially when the senior author is an accomplished electrophysiologist, there is no excuse for using ancillary effects of transduction and indirect arguments: the authors should in my estimation perform the necessary experiments.*

We have included direct measurements of the MET current in wild type and *mercury* mutants as requested. These measurements show that Tomt-deficient hair cells have no evoked MET current, consistent with the results of the FM dye uptake experiments and the previously published microphonic recordings.